# Principal component conditional generative adversarial networks for imbalanced ECG classification enhancement

**Chao Tang** [ID] *

School of Electronic Information Engineering, Changchun University of Science and Technology, Changchun, Jilin, China

* tangchao20250509@126.com

**Data availability statement:** This study utilizes publicly available medical datasets: the MIT-BIH Arrhythmia Database (https://physionet.org/content/mitdb/1.0.0/).

**Funding:** The author(s) received no specific funding for this work.

## Abstract

With over a century of development, electrocardiogram (ECG) diagnostics has become the preferred tool for healthcare professionals in cardiovascular disease diagnosis and monitoring. As wearable devices and mobile monitoring technologies become widespread, ECG data are trending toward diversity and long-term collection, making traditional manual annotation methods inadequate for massive data analysis demands. This research addresses core challenges in ECG signal classification—extremely imbalanced data, significant individual physiological differences, and difficulties in long sequence fitting—by proposing a Principal Component Analysis-based Conditional Generative Adversarial Network (PCA-CGAN). Through in-depth analysis of ECG signal principal component distribution characteristics, we discovered that just a few principal components can explain over 90% of signal variance, revealing the inherent inefficiency and limitations of traditional complete waveform generation methods. Based on this theoretical foundation, we shift the data augmentation paradigm from generating surface waveforms to generating high information density principal component features, resolving waveform jitter and heterogeneity issues present in traditional methods. Simultaneously, we designed a two-stage conditional encoding-decoding architecture that builds category-independent feature spaces from early training stages, fundamentally breaking the feature space bias caused by the "Matthew effect" and effectively preventing majority classes from compressing minority class features during generation. Using the Transformer's global attention mechanism, the model precisely captures key diagnostic features of various arrhythmias, maximizing inter-class differences while maintaining intra-class consistency. Experiments demonstrate that PCA-CGAN not only achieves stable convergence on a large-scale heterogeneous dataset comprising 43 patients for the first time but also resolves the "dilution effect" problem in data augmentation, avoiding the asymmetric phenomenon where Precision increases while Recall decreases. After

**Competing interests:** The authors have declared that no competing interests exist.

data augmentation, the ResNet model's average F1 score improved significantly, with particularly outstanding performance on rare categories such as atrial premature beats, far surpassing traditional methods like SigCWGAN and TD-GAN. This research redefines the objectives and methods of ECG signal generation from the theoretical perspectives of information entropy and feature manifolds, providing a systematic solution to data imbalance problems in the medical field while establishing a theoretical foundation for the application of ECG-assisted diagnostic systems in real clinical environments.

## Introduction

ECG diagnostics have been in clinical application since the beginning of the twentieth century, with over one hundred years of application history. To this day, ECG remains the preferred tool for healthcare professionals in the diagnosis and monitoring of cardiovascular diseases. ECG devices record voltage signals, allowing healthcare professionals to observe amplitude and phase changes to determine and identify various typical pathological states. Early ECG monitoring was primarily clinic-based, connecting supine patients to monitoring equipment through standard 12-lead electrodes. However, this monitoring form only scratched the surface of monitoring capabilities—patients could only be monitored while lying down, not only restricting patient activity but also requiring professional healthcare personnel to achieve proper monitoring, while simultaneously facing issues of collection costs and equipment accessibility.

With the development of modern medicine, medical devices have trended toward diversification and miniaturization. Wearable ECG monitoring devices and mobile ECG recorders have gradually become widespread, making long-term, continuous cardiac activity monitoring possible. These portable devices can collect ECG data during patients' daily activities, greatly expanding monitoring scenarios while significantly increasing data volume and complexity. Healthcare professionals face complex application contexts and analytical requirements, and medical staff using traditional manual annotation methods cannot solve such problems. Therefore, current ECG medicine seeks to introduce cybernetic algorithms to solve ECG classification problems through automated methods.

Before discussing the overall trends, we first introduce two key concepts.

1. **"Matthew Effect":** In machine learning training with imbalanced datasets, this refers to the phenomenon where majority classes with abundant data receive more comprehensive feature learning and parameter optimization, thus dominating the model, while minority classes with scarce data become further marginalized due to insufficient training samples. This creates a vicious cycle of "the rich get richer, the poor get poorer." In ECG signal classification, this effect manifests as majority classes like normal heart rate obtaining precise feature representations, while diagnostic features of rare classes like premature atrial contractions are ignored or distorted by the model.

2. **"Dilution Effect":** During data augmentation, this refers to the phenomenon where synthetic samples generated by traditional methods tend to reinforce the feature distribution of majority classes, further diluting and weakening the representation density of rare classes in the overall feature space. This results in augmented datasets that, while numerically balanced, remain biased toward majority classes in terms of feature quality, causing classifiers to improve in precision while their detection capability for rare classes actually decreases.

## Related work

In recent years, algorithms based on artificial intelligence have gradually become research focal points. Through automatic learning and fitting, these algorithms extract key patterns from ECG signals, greatly reducing manual costs while maintaining high accuracy and low latency.

### ECG classification methods

Fahad Khan et al. [1] used the classic deep ResNet structure for experiments on the MIT-BIH dataset, selecting five major heartbeat types for classification and achieving an average accuracy of 98.63%. Youhe Huang et al. [2] proposed a method using temporal features as a new representation input, maximizing differences between various inputs. Combined with attention mechanisms and CNN-LSTM models, this made classification methods more concise and efficient, achieving an average accuracy of 98.95% on MIT-BIH, effectively improving the model input section and achieving better results.

Xia Y et al. [3] discovered extreme metric results caused by data imbalance problems during experimentation and proposed a TCGAN model for generating ECG signals to alleviate data imbalance issues. However, their generation results only indirectly mitigated the problem; they enhanced four types of heartbeats and classified the enhanced results, achieving an overall accuracy of 94.69%, which to some extent alleviated the impact of the imbalance problem. Yang S et al. [4] combined different lead features from twelve-lead ECG summaries, effectively integrating different lead features through a multi-view method and using a multi-scale convolutional neural network structure to obtain temporal features at different scales, thereby enhancing network feature representation. Although achieving high effectiveness, this network structure places extremely high requirements on input, which is not conducive to the practical promotion of ECG classification detection.

Geng Q et al. [5] proposed an SE-ResNet structure for dynamic modeling of ECG feature sequences to obtain local and global information of signals, which was evaluated on the CSPC2018 dataset, achieving an average F1-Score of 82.7%. Qin K et al. [6] proposed an algorithm based on a novel biologically-inspired neuron operator, SelfONN, combining supervised and unsupervised training to obtain a pre-trained model for fine-tuning on various classification tasks, achieving an average AUC of 0.93 on the PTB-XL database. However, due to the excessive variety of heart rate data categories in the dataset, totaling 71 classes, the average metrics were not high compared to similar tasks, further highlighting the potential impact of data imbalance problems.

### ECG generation methods

Almost all the classification tasks above share a common problem: artificially inflated metrics due to data imbalance, which is exacerbated in multi-classification tasks. Delaney A M et al. [7] were the first to utilize a generative adversarial network architecture to generate cardiac rhythm patterns, with generated samples showing diversity. Subsequent researchers improved generation speed and quality from various aspects. Hazra D et al. [8] expanded the task, proposing a new generative adversarial network model, SynSigGAN, extending generation results to various biomedical signals with high correlation coefficients.

Subsequently, Zhu F et al. [9], considering the temporal features of ECG signals, proposed a BiLSTM-CNN GAN model that combined the advantages of LSTM and CNN, making GAN model training more stable. Chen J et al. [10]

proposed an ME-GAN to learn multi-view ECG generation methods, using existing heart disease conditions as panoramic ECG representations and splitting projections onto multiple views to produce ECG signals. Nguyen V D et al. [11] focused on fetal applications, using CGAN combined with time-frequency domain analysis to achieve fetal ECG extraction. Alcaraz J M L et al. effectively expanded ECG signals using diffusion models combined with structured state-space models. Adib E et al. discussed unconditional model and sinus heartbeat generation methods using improved denoising diffusion probabilistic waveforms. However, from an overall training perspective, both diffusion models and GANs experience significant waveform jitter in generation results, and diffusion models often perform worse than GANs in temporal tasks [12].

## Current limitations and challenges

In existing research, researchers have achieved a high performance and discussion to some extent, but there are still several major problems and limitations in the overall trend:

1. **Inefficient Paradigm of Complete Waveform Generation:** Almost all existing ECG domain data augmentation algorithms aim to generate complete waveforms. This approach ignores the characteristic that ECG signals contain large amounts of redundant information, consuming substantial computational resources to rebuild low information density regions during generation while blurring key pathological features. By analyzing the principal component distribution of the MIT-BIH dataset, we found that just a few principal components can explain over 90% of signal variance, indicating that traditional waveform generation methods are inherently inefficient and unable to precisely focus on features with true diagnostic value.

2. **Dataset Proportion Imbalance and the "Matthew Effect":** Due to the special nature of the medical field, patient disease types are increasingly finely divided, showing diversity characteristics. Patient disease records are encountered by chance rather than by design, and rare cases are greatly diluted in massive databases, thus being repeatedly diminished during model development and testing. Behind the high training metrics maintained by existing classification models lies the characteristic of artificially inflated metrics for common cases and insufficient detection of rare cases. Existing generation results can only indirectly alleviate the problem and do not fundamentally solve the "Matthew Effect" in generative model training—data-rich categories obtain more precise generation representations, further exacerbating imbalance.

3. **Individual Physiological Differences:** Existing ECG domain data augmentation algorithms are mostly based on generative models. Researchers use generative adversarial networks or diffusion models to capture ECG signal composition rules and learn patient ECG signal representations to expand datasets. However, physiological differences between patients increase the difficulty of learning ECG signals for generative models. Physiological differences between different patients are extreme, which is difficult to demonstrate in classification tasks but causes immense training difficulties in data augmentation tasks. Almost all existing research uses single-patient experiments or multi-patient combination training methods. Analyzing training results, their fitted data is patient-specific and lacks universality, causing models to learn many individual physiological differences irrelevant to classification tasks. When attempting to extend to multi-patient data, models face interference from high-variance features between patients, making it difficult to converge or produce meaningful samples.

4. **Long Sequence Fitting Problems:** ECG signals often span long recording times with numerous sampling points and rapid distortions. Current mainstream models perform poorly on long sequence tasks and struggle to capture distance dependency relationships among long sequence points. While attention mechanism-based models can theoretically handle long-distance dependency problems, their computational complexity grows quadratically with sequence length. A single-cycle heart rate signal consists of 300–360 sampling points, while typically only a few sampling points are

needed to explain waveform variance. Current data augmentation tasks all aim to generate complete waveforms; therefore, for classification tasks, cardiac rhythm patterns contain many irrelevant redundant points that obscure key pathological features.

## Methodology and technical contributions

### Ethics statement

This study utilized a fully anonymized public dataset (MIT-BIH Arrhythmia Database), which has been widely used in research and received appropriate ethical clearance by its original collectors. According to the policies of our institution, secondary analysis of such publicly available anonymized datasets that have already received ethical approval does not require additional ethical review.

In response to existing issues of data imbalance, individual differences, and long sequence fitting problems in the ECG signal classification field, this research proposes an innovative solution that breaks through the limitations of traditional methods from multiple dimensions. Our work includes the following innovations:

1. **Paradigm Shift in ECG Signal Representation:** We are the first to propose a new method that transforms data augmentation from generating complete waveforms to generating principal component features. Through principal component analysis of the MIT-BIH dataset, we discovered that the first 7 principal components can explain over 90% of signal variance, and different types of heart rates exhibit distinct clustering characteristics in principal component space. This finding prompted us to reconsider the essential problem of ECG signal enhancement: what truly has clinical value are the key diagnostic features in the signal, not the abundant redundant information in complete waveforms. By generating principal component features with higher information density, we effectively avoid the waveform jitter problems in traditional methods while bypassing learning barriers caused by physiological differences between individuals. This enables the model to focus on more representative heart rate principal components, essentially transforming "morphologically similar" waveform generation into "conceptually similar" feature generation.

2. **Transformer-based Conditional Generative Adversarial Network:** To solve the extreme imbalance problem in ECG datasets, we designed a novel conditional generative adversarial network architecture. Its core innovation lies in using the Transformer's global attention mechanism to replace traditional CNN/RNN structures. Experiments proved that the global attention mechanism captures feature dependency relationships more effectively of various arrhythmias, ensuring that generated samples retain category-specific diagnostic features. Unlike traditional GANs' instability when training on imbalanced data, by precisely modeling the spatial distribution of various ECG features, our method can specifically enhance key features of rare categories, effectively alleviating classification bias caused by data imbalance. This allows classification algorithms to maintain high classification accuracy while significantly reducing artificially inflated metrics for common cases.

3. **Two-Stage Conditional Encoding-Decoding Strategy:** We designed a two-stage training conditional encoding-decoding architecture. In the first stage, the conditional encoder embeds category information into feature representations, enabling the model to build independent feature spaces for different categories from the beginning of training. This strategy fundamentally solves the feature space bias problem caused by the "Matthew Effect" in traditional methods, preventing feature space compression of minority classes by majority classes during generation. In the second stage, the pre-trained encoder-decoder combines with the conditional GAN to achieve precise generation of ECG signal principal components. This architectural design not only makes model training more stable but also greatly reduces learning complexity. By reducing useless computations, the model can concentrate its limited expressive capacity on feature dimensions with the most diagnostic value. Compared to traditional methods, our approach achieves higher quality sample generation while reducing computational resource consumption.

4. **Resolving the Dilution Effect in Data Augmentation:** We deeply analyzed the "dilution effect" problem commonly present in ECG signal data augmentation—samples generated by traditional methods often strengthen majority class features, further diluting rare category information, leading to the asymmetric phenomenon of increased classifier Precision but decreased Recall. Through T-SNE visualization analysis, we found that this dilution effect stems from biased learning of generative models in feature space, causing minority class samples to be further marginalized. Our PCA-CGAN fundamentally avoids this problem through three mechanisms: first, the principal component generation strategy focuses the model on core features that distinguish categories rather than surface waveforms easily affected by individual differences; second, conditional encoding ensures each category has an independent representation area in feature space, preventing majority classes from dominating the feature space; third, the Transformer architecture's global attention enables the model to precisely capture subtle differences between categories, maximizing inter-class differences while maintaining intra-class consistency. Experiments prove that our method successfully avoids the asymmetric performance problem of classifiers after expansion, dramatically improving Class A Recall from 79.58% to 99.56% while maintaining a high Precision of 99.86%, achieving a win-win situation for both precision and sensitivity.

5. **Breaking the Single-Patient Limitation to Achieve Multi-Patient ECG Signal Generation:** This research successfully overcomes the long-standing "single-patient experiment" limitation in the ECG data augmentation field, achieving stable training on a large-scale heterogeneous dataset containing 43 patients for the first time. Traditional methods face two core challenges when extended to multi-patient scenarios: first, physiological differences between patients make it difficult for models to learn unified feature distributions; second, rare categories are further diluted in multi-patient data. Our K-fold cross-validation experiments show that PCA-CGAN achieved stable convergence on data from 43 patients, with an average NPRD of 30.6212 and a standard deviation of only 2.4101, breaking the limitation that existing generative models can only be trained on single-patient or few-patient data. In-depth analysis indicates that this breakthrough stems from our principal component generation strategy fundamentally changing the feature learning objective—from learning highly individualized waveform appearances to capturing cross-patient pathological principal component patterns. The complete waveforms that traditional methods attempt to learn contain many individual-specific features, leading to high variance between different patients' data, while principal component representations distill the essential features of diseases, forming low-dimensional, high information density representations that greatly reduce the difficulty of cross-patient learning. This innovation not only improves the generalization ability of generative models but, more importantly, removes key barriers to the application of ECG-assisted diagnostic systems in real clinical environments, enabling systems to learn from diverse patient populations, adapt to physiological differences between individuals, and maintain reliable performance on previously unseen new patient data.

In summary, this research starts from the essence of ECG signals to reconsider and redefine the goals and methods of ECG data augmentation, breaking through the bottlenecks of traditional methods on extremely imbalanced data and providing an efficient, stable solution with clinical practical value for ECG classification tasks. Through innovative principal component conditional generation strategies, we not only solve technical challenges but also provide new possibilities for precision and popularization of clinical ECG diagnosis.

## Data preprocessing and feature analysis

In this research, we adopted the widely used MIT-BIH dataset as our experimental data source [13]. The database includes 48 half-hour long two-channel ECG recordings collected from 47 patients (including 25 males and 22 females, age range 23 to 89 years), with a sampling frequency of 360 Hz. Each sampling point is recorded using 11-bit resolution quantization data. The dataset primarily contains five beat types, which can be further subdivided into ten types, with rich and diverse annotations. Therefore, the MIT-BIH dataset has become a representative imbalanced dataset [14]. We selected five representative types of cardiac rhythm patterns as our experimental subjects: Normal beat (N), Ventricular

premature beat (V), Atrial premature beat (A), Left bundle branch block (L), and Right bundle branch block (R). The proportion of each type in the dataset is shown in Fig 1, where normal heart rate accounts for 77% [15], while abnormal heart rates together only account for 1/4 of the total. Taking the atrial premature beat (A) category as an example [16], it only accounts for 2.114% in the MIT-BIH dataset. This severe data imbalance phenomenon reflects actual clinical situations, and the extremely unbalanced distribution causes classification algorithms to tend to classify samples as the majority class, leading to insufficient recognition of minority classes.

We selected wavelet transform as our preprocessing method [17]. As a typical temporal periodic signal, cardiac rhythm patterns exhibit rapid amplitude distortion and periodic characteristics. Therefore, common Fourier transforms cannot effectively filter the results. We plotted the power spectral density diagrams for the five types of ECG signals, as shown in Fig 2. The five categories of ECG signals show significant similarities in spectral features, precisely illustrating that relying solely on Fourier transform makes it difficult to effectively distinguish between different categories of ECG signals.

Furthermore, as shown in Fig 3, when we observe the time-domain average waveforms of the five types of ECG signals, we can find obvious morphological differences between them. These time-domain features provide rich distinguishing information for different categories of ECG signals, which frequency-domain analysis cannot fully capture.

Wavelet transform provides both time-domain and frequency-domain analysis windows simultaneously. Through wavelet basis functions of different scales, it can precisely locate the time-domain features of signals at different frequencies, as shown in Equation 1.

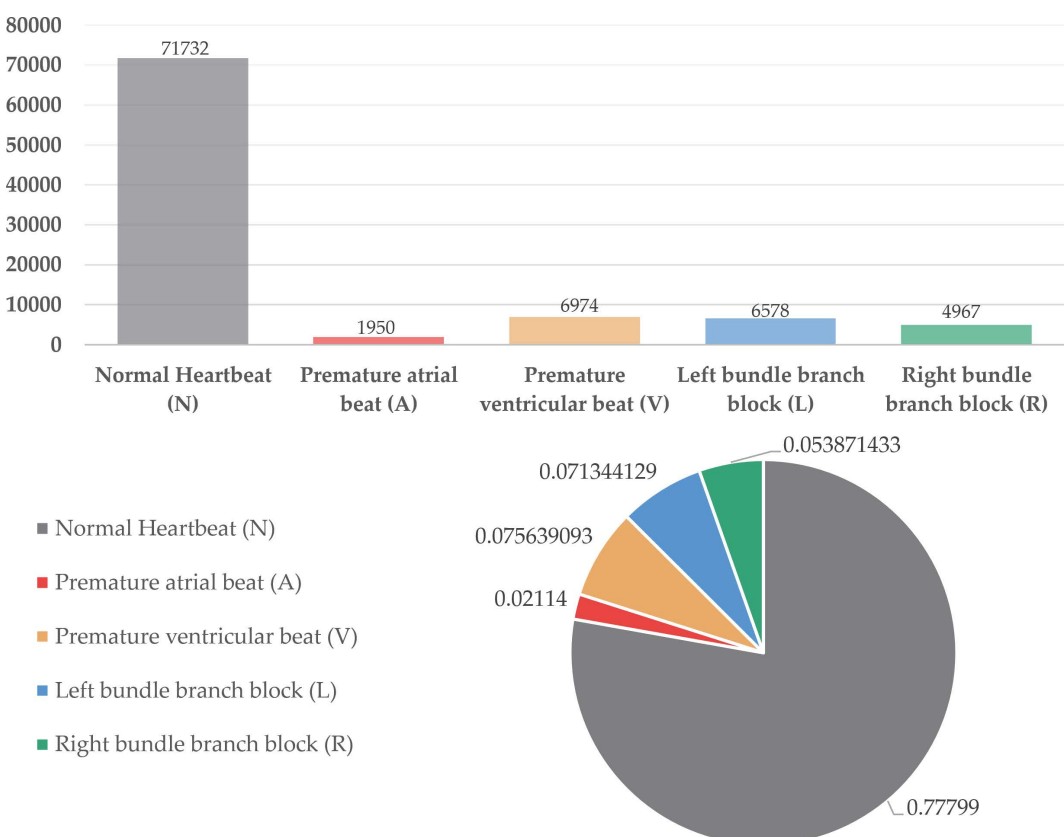

**Fig 1. MIT-BIH dataset proportion analysis chart.**

$$W(a, b) = \int_{-\infty}^{\infty} x(t) \cdot \frac{1}{\sqrt{a}} \psi(\frac{t-b}{a}) dt$$

(Eq.1)

In this context, $x(t)$ represents the original signal, $\psi()$ is the wavelet basis function, $a$ is the scale parameter, $b$ is the translation parameter, and $W(a, b)$ is the wavelet coefficient. In practical applications, we employ the discrete wavelet transform DWT, which can be expressed as shown in Equation 2.

$$DWT(j, k) = 2^{-j/2} \int_{-\infty}^{\infty} x(t)\psi(2^{-j}t - k) dt$$

(Eq.2)

Compared to Fourier transform, wavelet transform uses oscillating waveforms of finite length as basis functions, which better adapt to the local characteristics of non-stationary signals. We tested various wavelet bases to determine the most effective combination. Our experiments included the classical Daubechies, Symlets, Coiflets, and Haar wavelet families. The test results for different wavelet families are presented in Table 1. Based on these results, we selected the optimal combination of Daubechies wavelet family with db7 and 9-level decomposition to achieve the best preprocessing configuration. The filtering results are shown in Fig 4.

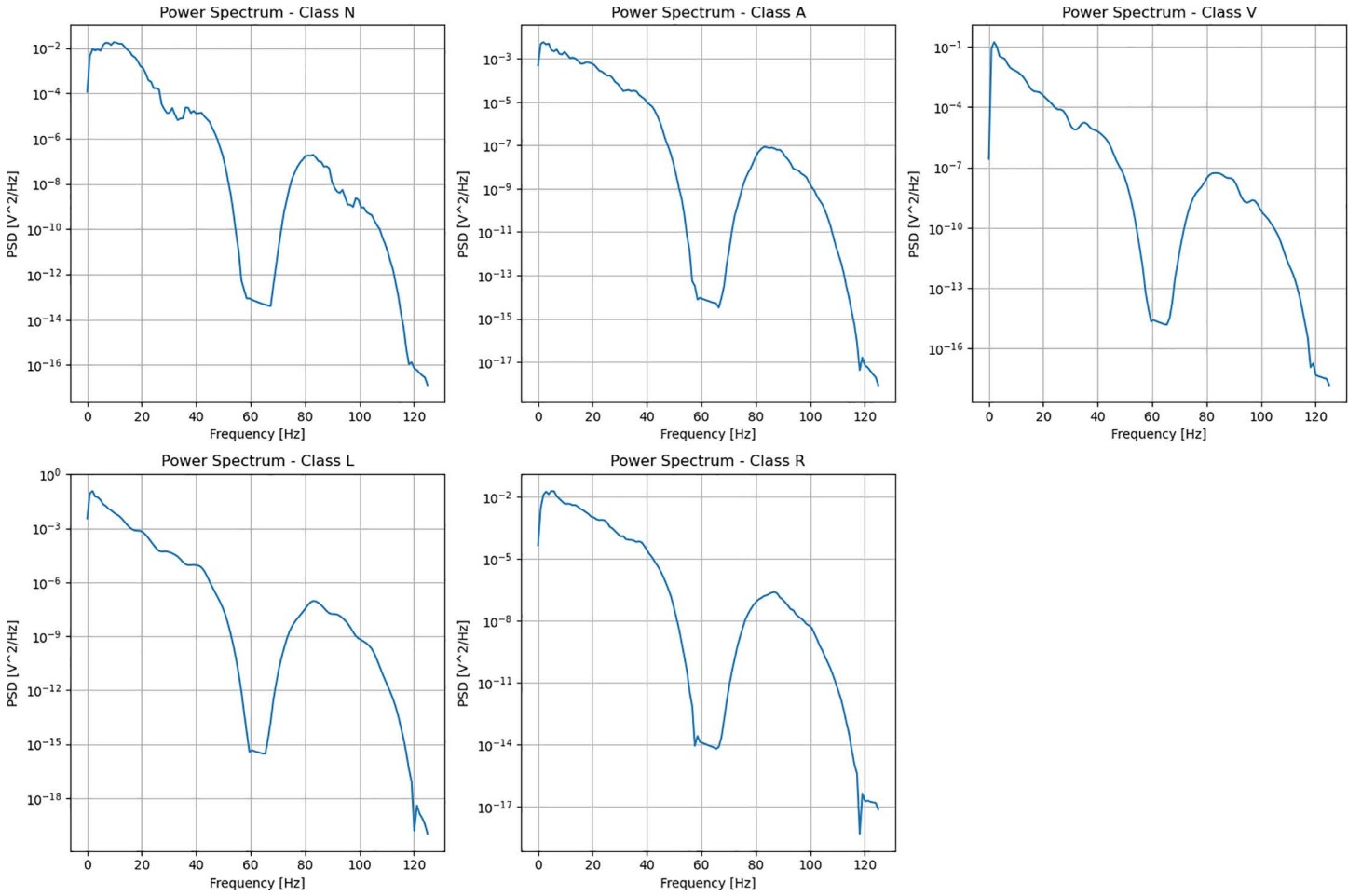

**Fig 2. Power spectral density comparison chart of various heart rate types in the MIT-BIH dataset.**

In heart rate classification algorithms, some researchers have adopted PCA dimensionality reduction methods commonly used in epilepsy studies [18]. By preserving the main components of signals while eliminating numerous irrelevant points, classification efficiency and accuracy can be improved. We applied PCA dimensionality reduction to ECG signals to analyze variance explanation, with results shown in Fig 5. For a single-cycle ECG signal, the first principal component alone explains approximately 60% of variance [19], while the first 7 principal components together explain 90% of total variance [20]. This indicates we can effectively reduce data dimensionality while preserving most signal information

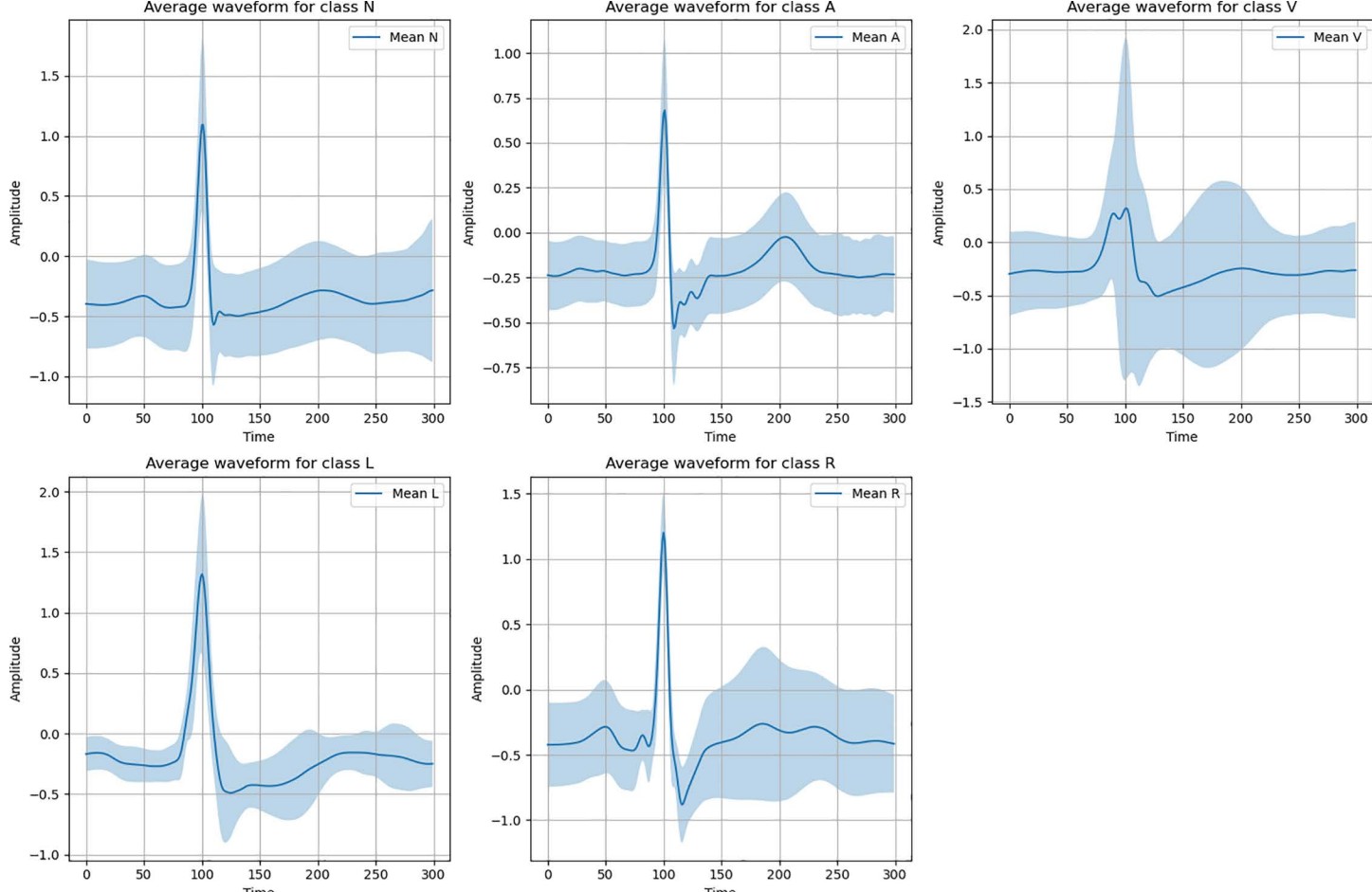

**Fig 3. Average amplitude comparison chart of various heart rate types in the MIT-BIH dataset.**

**Table 1. Comparison of denoising effects of ECG signals with different wavelet parameters.**

| Wavelet Type | Average SNR (dB) | Average R-wave Peak Retention Rate | Average RMSE | Correlation Coefficient |
|---|---|---|---|---|
| Daubechies | **30.4967** | **100.00** | **0.01068** | **0.9983** |
| Symlets | 30.4703 | 99.01 | 0.01073 | 0.9983 |
| Coiflets | 30.4967 | 100.00 | 0.01068 | 0.9981 |
| Haar | 27.6039 | 100.00 | 0.01501 | 0.9968 |

by retaining few principal components, demonstrating that very few points can represent the dominant patterns in ECG signals.

Therefore, many existing ECG studies generate complete waveforms to supplement datasets, aiming to improve classification algorithm metrics by training models to recognize principal component changes in various rare waveforms [21]. However, we believe this approach actually hinders the model's ability to identify rare signals. Data augmentation tasks waste resources by focusing on generating irrelevant points. Learning surface features rather than principal component features often causes models to excessively focus on waveform appearances rather than essential structures [22], potentially masking the most diagnostically valuable features. Complete waveform generation introduces redundant information that increases computational complexity, significantly affects processing speed, and makes it difficult for models to focus on true pathological markers [23].

Many pathological features in ECG signals manifest as specific changes in particular wave segments, which can be effectively captured by the first few principal components. Since principal components are already optimized feature representations, directly generating these components allows models to learn effective classification boundaries more quickly, reducing training time on redundant features. Providing high-information-density principal component features theoretically enables more accurate simulation of core feature changes under various pathological conditions, avoiding distortions that may occur in complete waveform generation.

We also performed T-SNE dimensionality reduction on heart rate waveforms to observe signal distribution patterns, as shown in Fig 6. Different ECG signal types form distinct clusters in two-dimensional space, providing an intuitive perspective for understanding similarities and differences between various signals. We retained 16 principal component sampling

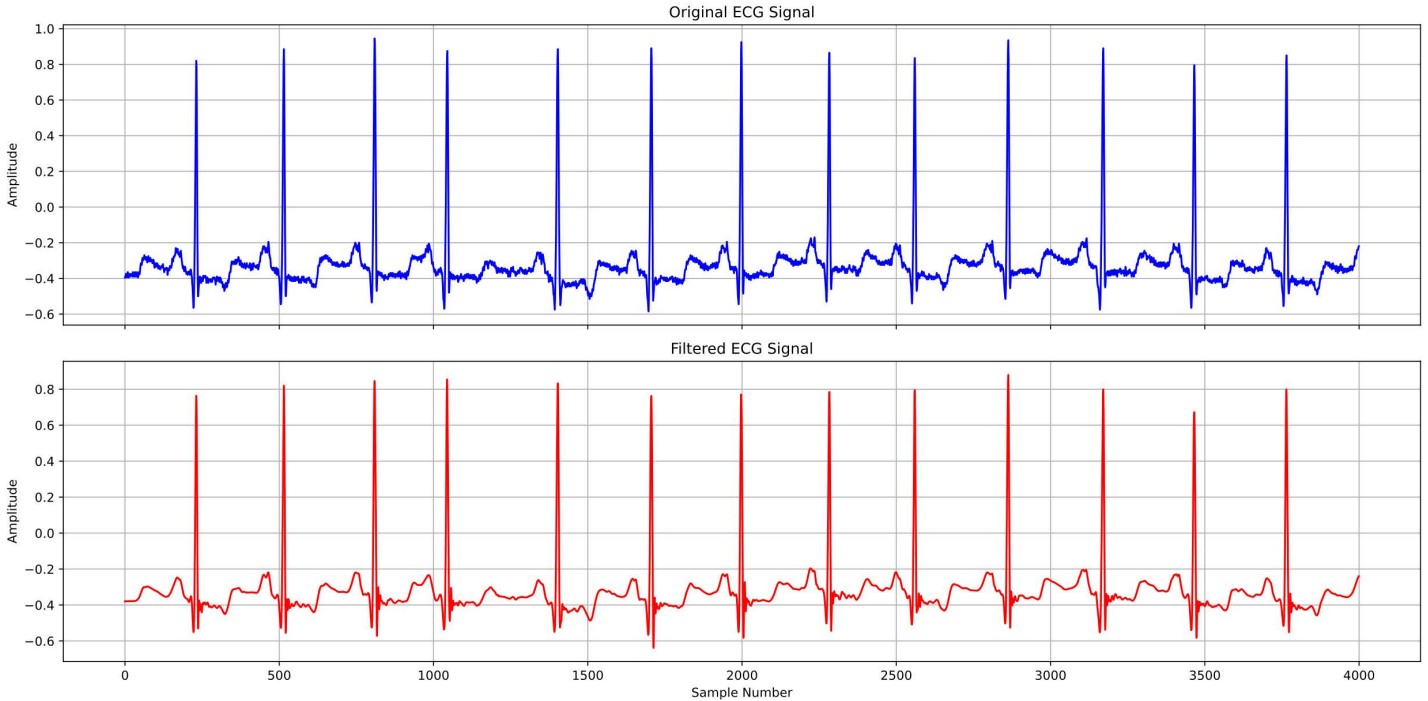

**Fig 4. Wavelet transform preprocessing results.**

points from PCA dimensionality reduction and combined them into a matrix visualized as a heat map to observe differences between principal component types, as shown in Fig 7. By comparing heat map patterns across categories, feature differences among various ECG signals in principal component space become clearly visible.

Based on this analysis, we shifted our goal from generating surface feature signals to generating principal component features of various heart rate signal types. By directly generating high-information-density principal component features and combining techniques such as conditional embedding, we can specifically enhance key features of rare categories rather than simply copying or slightly varying existing samples.

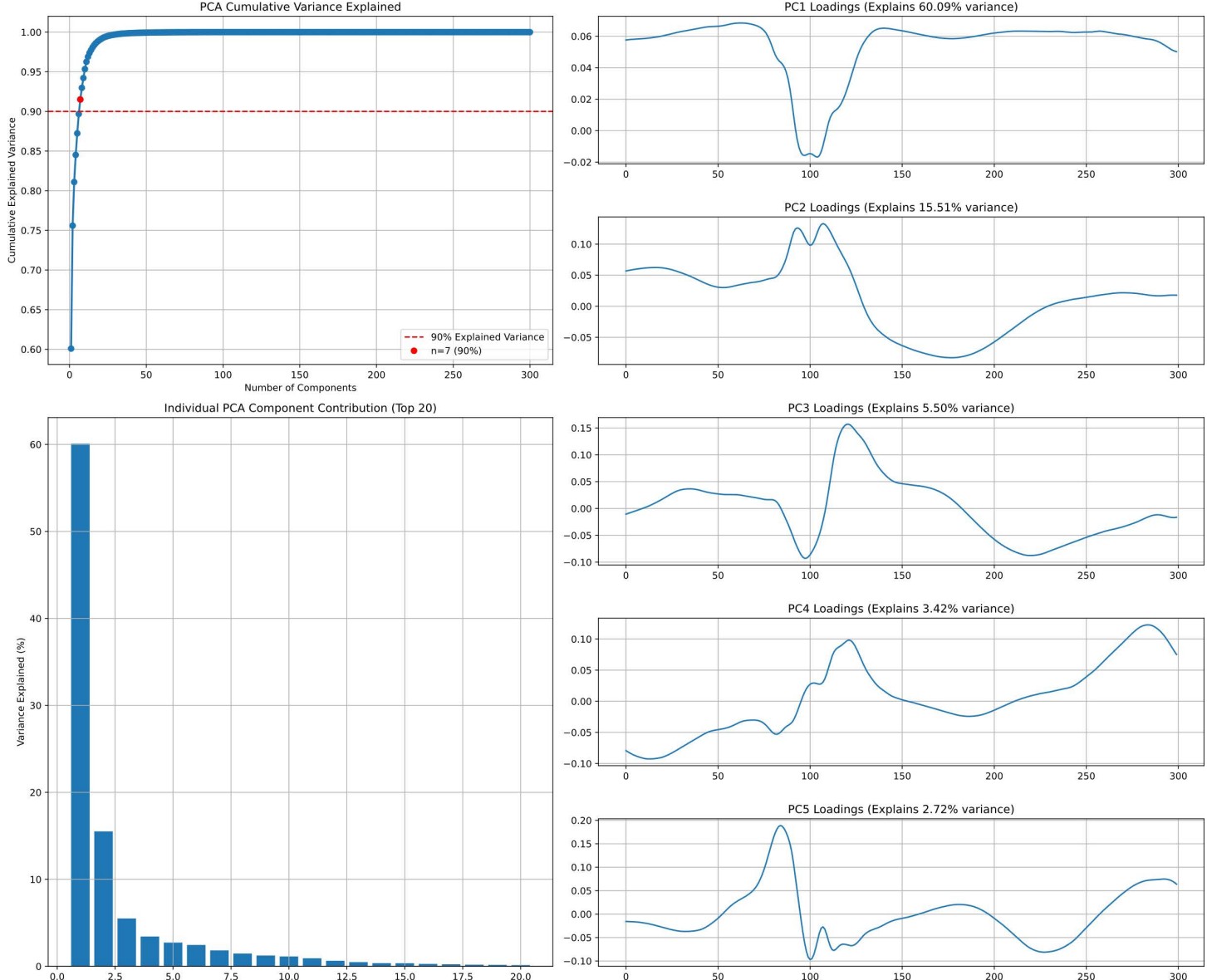

**Fig 5. PCA principal component analysis of MIT-BIH Dataset.**

## Baseline algorithm performance evaluation

We selected two representative classification algorithms: Random Forest from the machine learning domain and ResNet from deep learning, to represent the common issues of artificially inflated metrics and imbalance problems in existing classification approaches. The structural diagrams of both algorithms are shown in Fig 8.

We conducted classification experiments using the imbalanced MIT-BIH dataset described earlier with both algorithms, incorporating PCA dimensionality reduction as a comparative experiment. This demonstrates both the artificially high metrics of existing classification algorithms and verifies the effectiveness of PCA principal component analysis. The classification results for all four scenarios are listed in Table 2.

First, both ResNet and RF models show minimal differences after incorporating PCA dimensionality reduction compared to the original models, while computation time increases exponentially. As shown in Fig 9, models using PCA technology capture efficiency faster, achieve higher metrics, and reach convergence earlier. The models improve capture ability while reducing computational load, which proves our earlier assertion that generating complete heart rate waveforms to improve classification algorithms consumes excessive computational resources with minimal efficiency. Moreover, the final model metrics are not ideal, as incorporating numerous irrelevant points affects model fitting ability, further supporting our argument that generative models should aim to generate principal components rather than complete waveforms when improving classification algorithm performance.

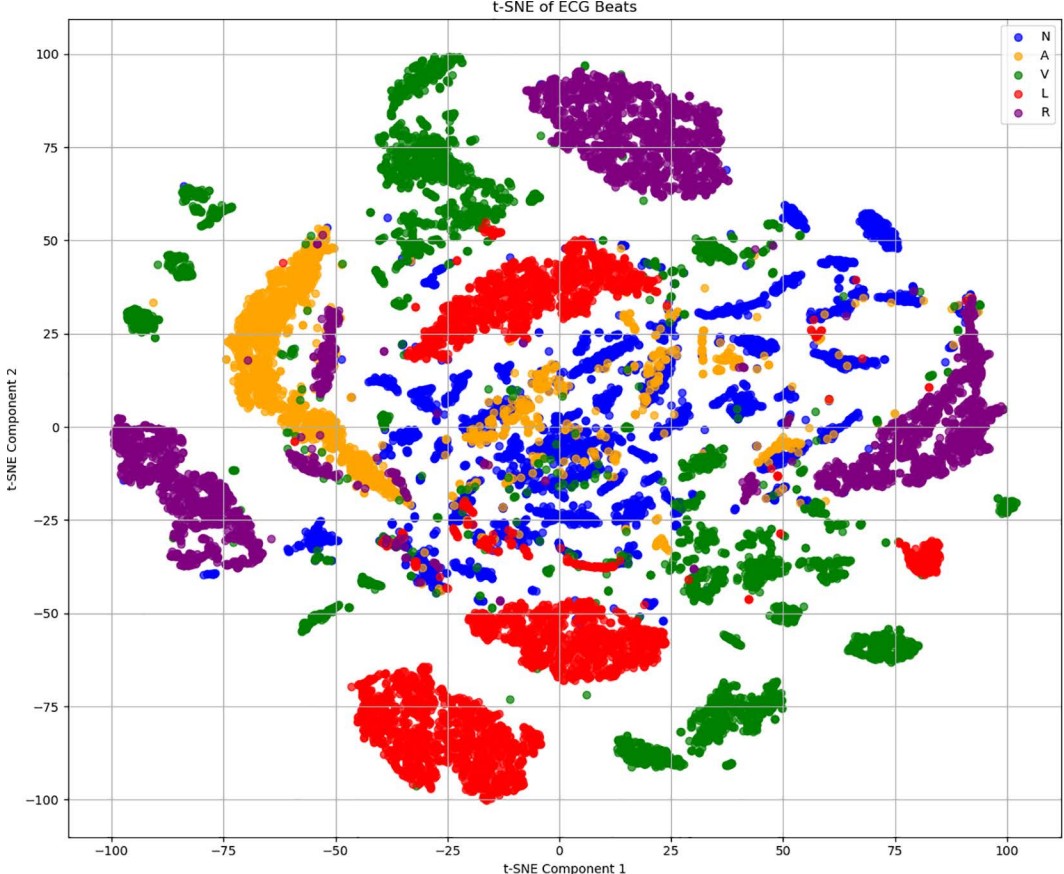

**Fig 6. T-SNE dimensionality reduction visualization of MIT-BIH Dataset.**

Heatmaps of ECG Data by Class (5 samples per class)

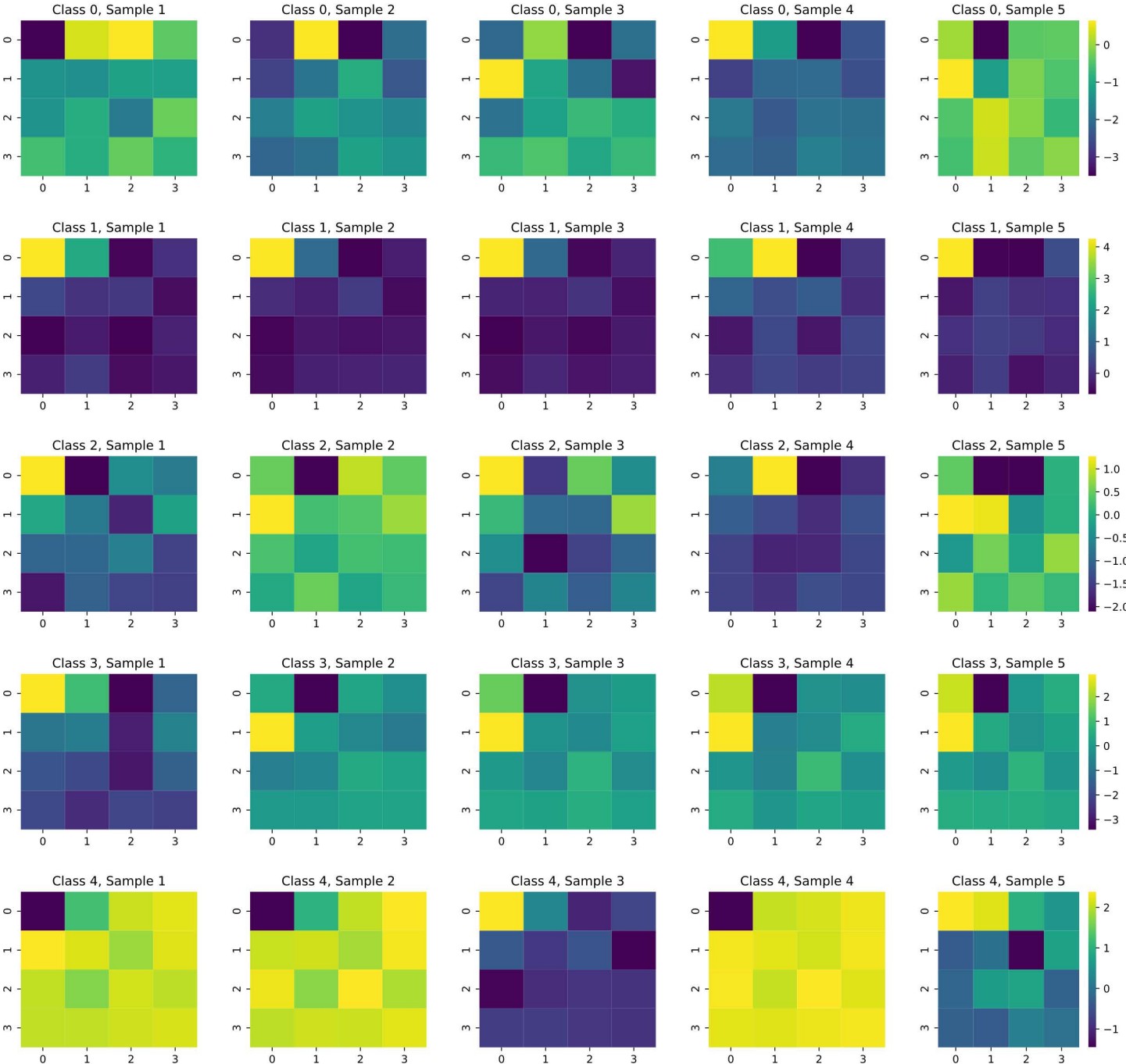

**Fig 7. Comparison of Principal Component Heat Maps for Different Heart Rate Signal Types.**

Second, as shown in Fig 10, we plotted confusion matrices for all four models. Most existing research only uses Accuracy for model comparison, but Table 2 clearly shows that models have common problems: while Accuracy reaches relatively high levels, other metrics remain relatively low. Therefore, we present test results for various heart rate types combining ResNet and RF with PCA in Table 3, and different PCA dimensionality reduction results in Table 4.

Analyzing Tables 2–4 together reveals several key characteristics:

1. Whether combined with PCA dimensionality reduction or not, macro-average Recall values consistently remain low while macro-average Precision values are relatively high. This results from dataset imbalance, causing models to bias parameters toward majority classes during learning, resulting in insufficient learning for rare categories. This is observable in the confusion matrices in Fig 10, where numerous rare class samples are misclassified as normal heart rate—which would have serious consequences in medical tasks. Therefore, introducing data augmentation methods is necessary.

2. When combining classification algorithms with PCA dimensionality reduction, increasing principal component points causes model metrics to first increase then decrease, as shown in Fig 11. Overall, macro-average metrics follow this pattern, achieving optimal performance at approximately 36 sampling points while maintaining computational efficiency. With further increases, metrics plateau or slightly decrease. This phenomenon affects machine learning algorithms

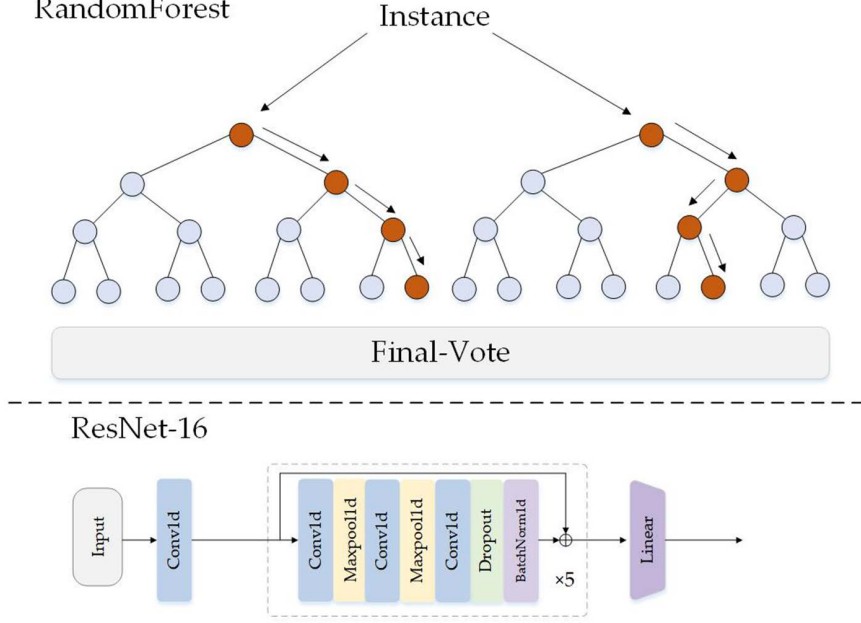

**Fig 8. Structural Diagrams of RF Model and ResNet Model.**

**Table 2. Comparison of Model Five-classification Results.**

| Model | Acc | AUROC | Precision | Recall | F1-score | Time count(s) |
|---|---|---|---|---|---|---|
| ResNet (PCA component = 32) | 98.52 | 96.72 | 99.27 | 94.25 | 96.55 | 229.082 |
| ResNet(without PCA) | 97.65 | 95.55 | 97.23 | 92.20 | 94.54 | 478.085 |
| RF(PCA component = 32) | 96.81 | 99.36 | 98.24 | 87.01 | 91.87 | 4.906 |
| RF(without PCA) | 97.21 | 98.96 | 95.99 | 89.20 | 92.38 | 15.564 |

more significantly than deep learning models, confirming our analysis that excessive irrelevant points not only impact computational efficiency but also hinder feature extraction, meaning that merely generating surface characteristics can impede the feature fitting process of classification algorithms.

3. Corresponding to our earlier PCA explained variance analysis, the first 10 principal component points explain the vast majority of variance. Combined with Fig 11 findings, this verifies our analysis that existing heart rate domain data augmentation models focused on generating complete waveforms to improve classification metrics have fundamental limitations, and the goal should instead be generating principal components of various heart rate types.

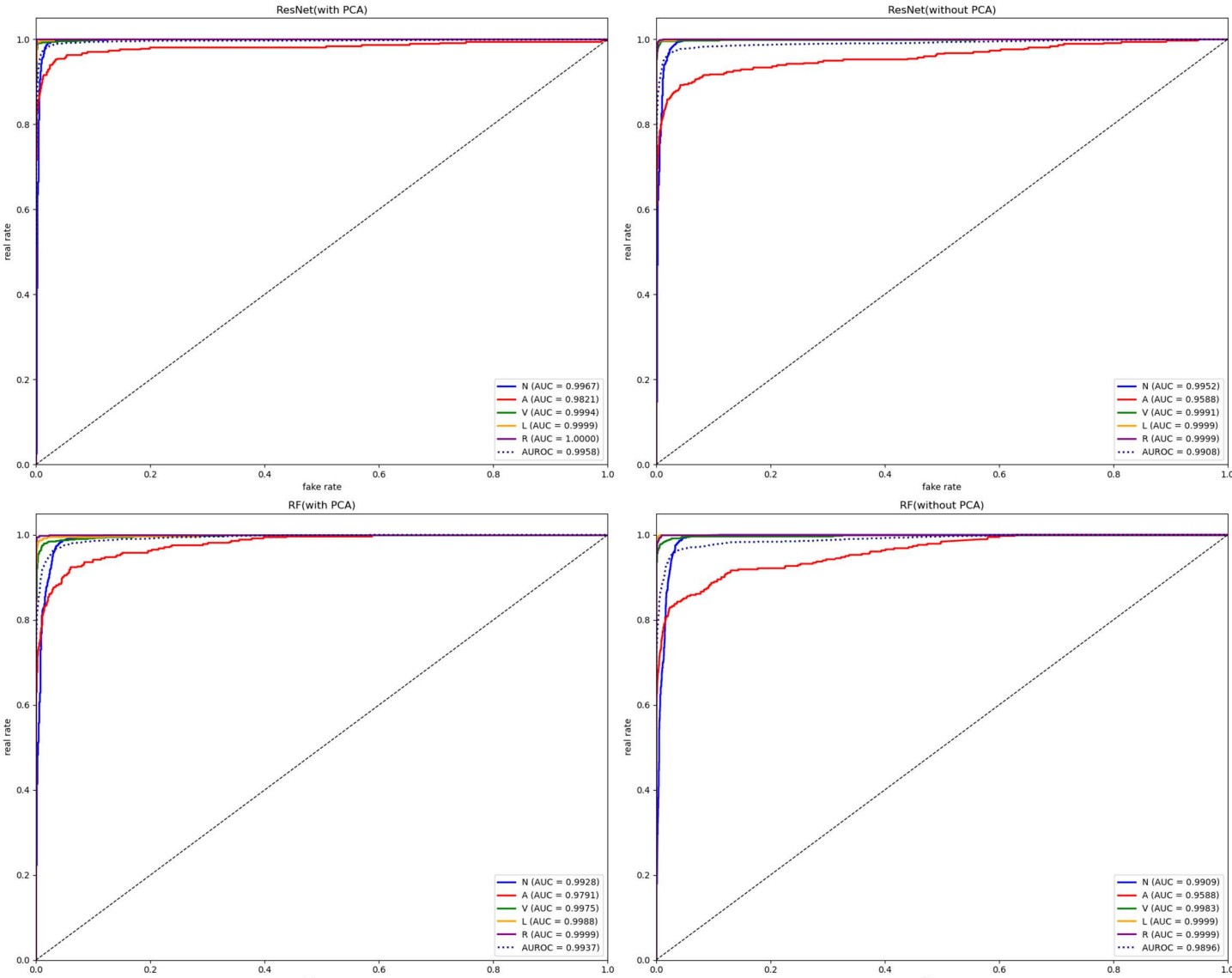

**Fig 9. AUROC Graphs of RF Model and ResNet.**

## Network architecture design

Based on the preceding analysis, as shown in Fig 12, we propose a novel conditional generative architecture called PCA-CGAN. Through a two-stage training process, this model achieves more reliable generation results. The PCA-CGAN model consists of four components: Conditional Encoder, Conditional Decoder, Conditional Generator, and Conditional Discriminator, all implemented using Transformer architecture.

The model training is divided into two stages. In the first stage, input ECG signals undergo PCA dimensionality reduction, with the retained principal components and their heart rate type serving as conditional input to the Conditional

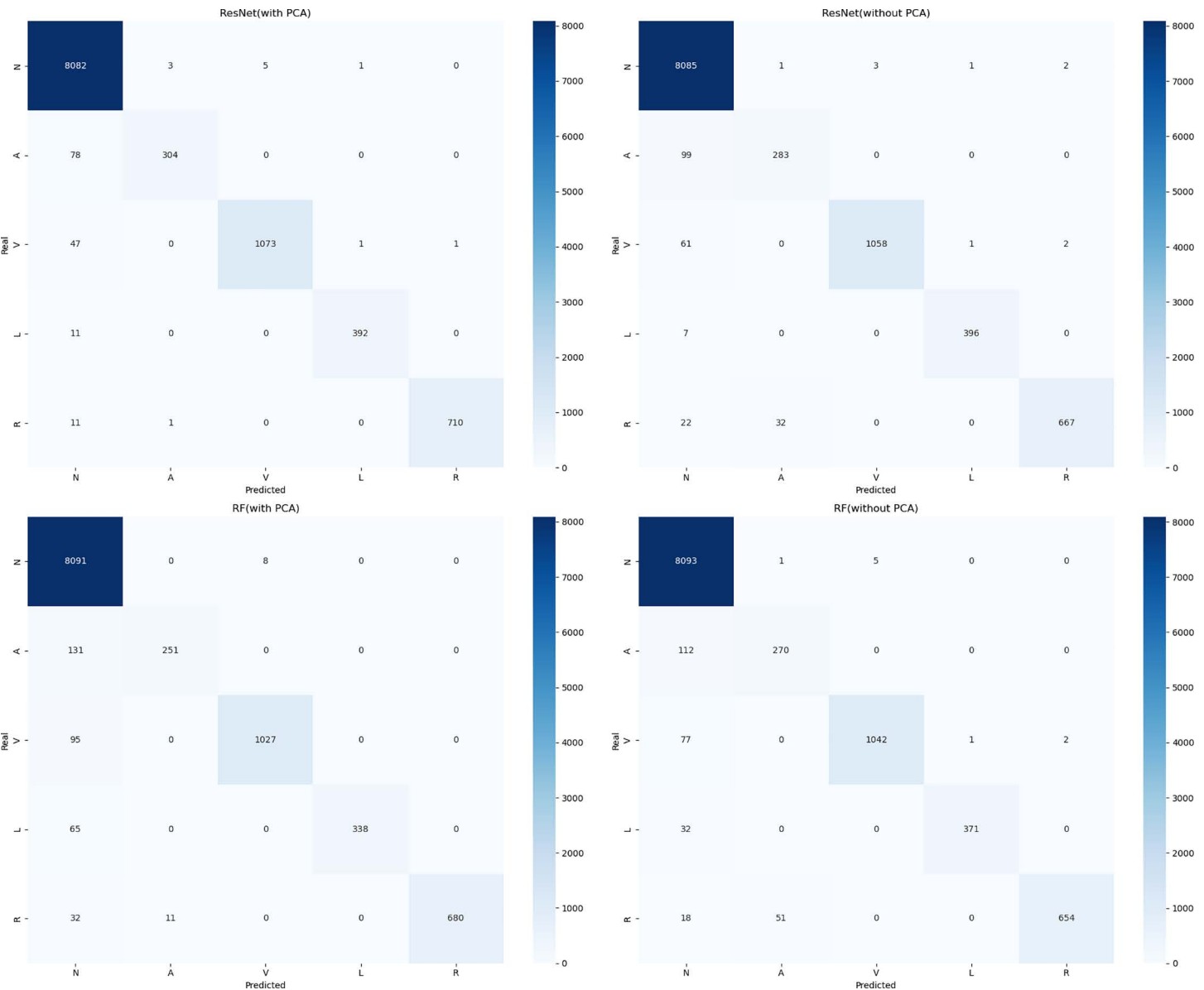

**Fig 10. Confusion Matrix Graphs of RF Model and ResNet Model.**

Encoder, then output by the Conditional Decoder. This structure maximizes model efficiency [24]. After PCA dimensionality reduction, the cardiac rhythm patterns have numerous irrelevant points filtered out, preserving the highest information content feature matrix. We visualize this process in Fig 13. Simultaneously, the conditional encoder embeds category information into the feature representation, enabling the model to more precisely learn feature distributions of various heart rate signal types [7]. We visually differentiate this from traditional Enc-Dec architecture, avoiding potential imbalance issues from the beginning of model training. The Conditional Encoder-Decoder uses Transformer architecture, leveraging its powerful attention mechanism to effectively capture dependencies between principal components [25], overcoming limitations of traditional models when processing sequence data. The combination of global attention and principal component matrices maximizes model capture efficiency, avoiding various effects caused by long-distance dependencies.

In the second training stage, we integrate the pre-trained encoder-decoder into a conditional generative adversarial network framework. The Conditional Generator receives random noise and heart rate type conditions as input, generating principal component representations of corresponding ECG signal categories rather than complete waveforms. The Conditional Discriminator determines whether the generated principal components are authentic and belong to the correct category. This design enables the model to specifically enhance key feature representations of rare categories in the dataset rather than simply copying existing samples. Compared to traditional GANs, our PCA-CGAN significantly reduces

**Table 3. Test Results of Various Heart Rate Types Combining ResNet and RF with PCA.**

| Model | Type | Precision | Recall | F1-score |
|---|---|---|---|---|
| ResNet(with PCA) | N | 98.21 | 99.89 | 99.04 |
| | A | 98.70 | 79.58 | 88.12 |
| | V | 99.54 | 95.63 | 97.55 |
| | L | 99.49 | 97.27 | 98.37 |
| | R | 99.86 | 98.34 | 99.09 |
| RF(with PCA) | N | 96.16 | 99.90 | 98.00 |
| | A | 95.80 | 65.71 | 77.95 |
| | V | 99.23 | 91.53 | 95.22 |
| | L | 100.00 | 83.87 | 91.23 |
| | R | 100.00 | 94.05 | 96.94 |

**Table 4. Comparison of Different PCA Dimensionality Reduction Results for ResNet and RF.**

| Model | PCA Component | Total Acc | Macro-Average Precision | Macro-Average Recall | Macro-Average F1-score |
|---|---|---|---|---|---|
| ResNet | 9 | 97.93 | 96.84 | 92.04 | 94.24 |
| | 16 | 98.53 | 99.13 | 94.20 | 96.44 |
| | 25 | 98.59 | 99.11 | 94.62 | 96.70 |
| | 36 | 98.77 | 98.96 | 95.46 | 96.98 |
| | 49 | 98.70 | 99.17 | 95.01 | 96.94 |
| | 64 | 98.59 | 99.10 | 94.59 | 96.68 |
| RF | 9 | 96.19 | 93.84 | 85.93 | 89.59 |
| | 16 | 96.98 | 97.89 | 87.77 | 92.20 |
| | 25 | 97.06 | 98.49 | 88.05 | 92.60 |
| | 36 | 96.78 | 98.32 | 87.00 | 91.87 |
| | 49 | 96.38 | 98.01 | 85.42 | 90.80 |
| | 64 | 96.57 | 97.87 | 86.24 | 91.28 |

training difficulty and computational resource consumption by generating higher information density principal component features, while avoiding common waveform jitter problems in complete waveform generation.

Furthermore, we have written the model pseudocode as shown in Table 5.

## Algorithm workflow and implementation

The PCA-CGAN model operates according to the workflow shown in Fig 14. The process begins with input of original ECG signal data and corresponding category labels. The ECG signal data then enters the PCA downsampling module, where wavelet transform is applied for signal denoising, followed by principal component analysis to reduce data dimensionality while preserving the most informative features of the ECG signal. The downsampled PCA principal components and category labels are jointly input to the conditional encoding module, which embeds category information into feature representations, enabling the model to distinguish specific categories from the beginning of training. By establishing unique feature spaces for each heart rhythm type, this helps resolve class imbalance issues.

The dimensionally reduced and encoded representations after PCA principal component analysis are subsequently trained in a Transformer-based architecture. Ideally, the model learns stable mapping relationships between latent space and PCA principal component representations while maintaining category-specific information. The trained representations are then processed through the conditional decoding module, which preserves category-specific features embedded

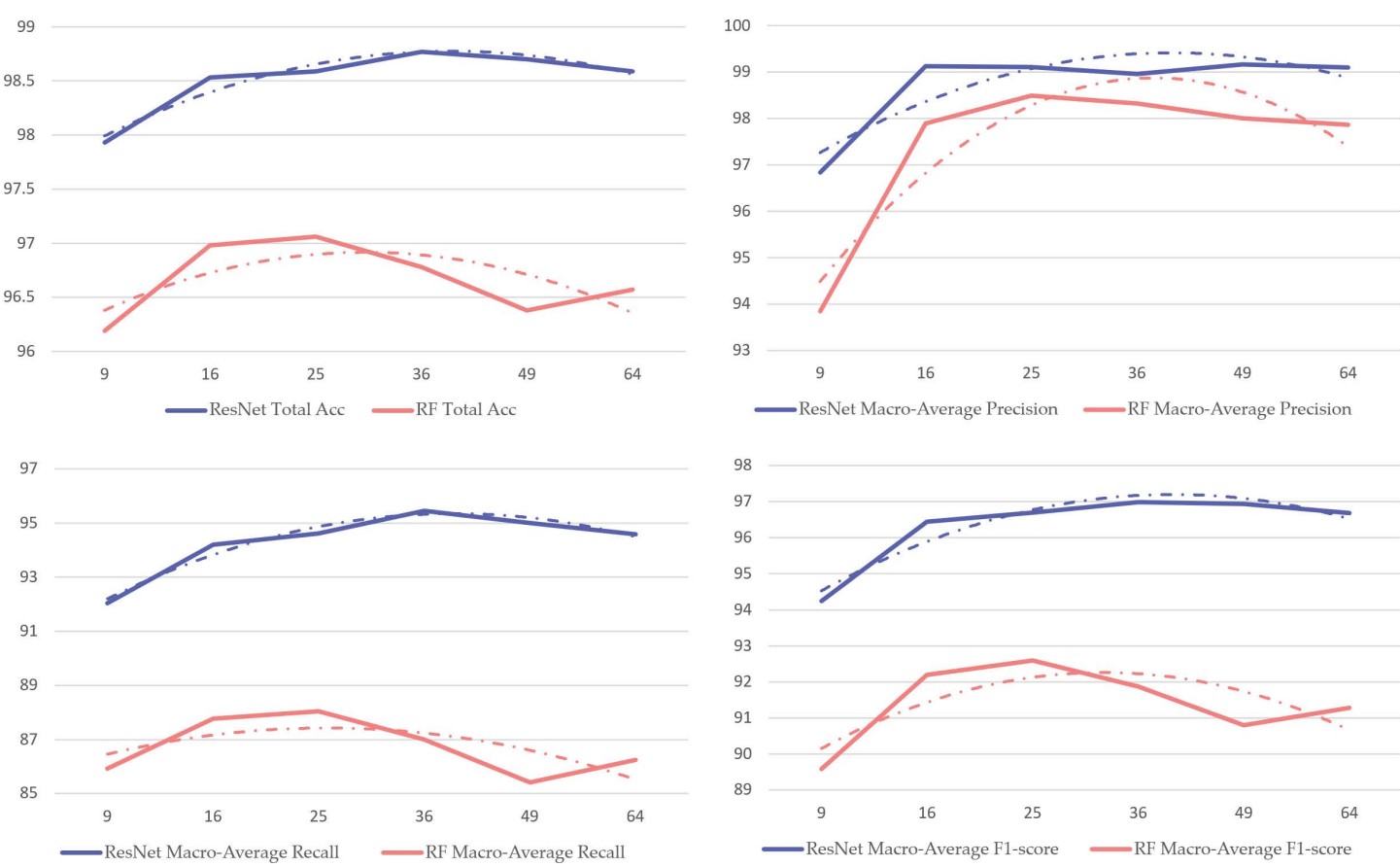

**Fig 11. Changes in Metrics with Number of PCA Principal Components.**

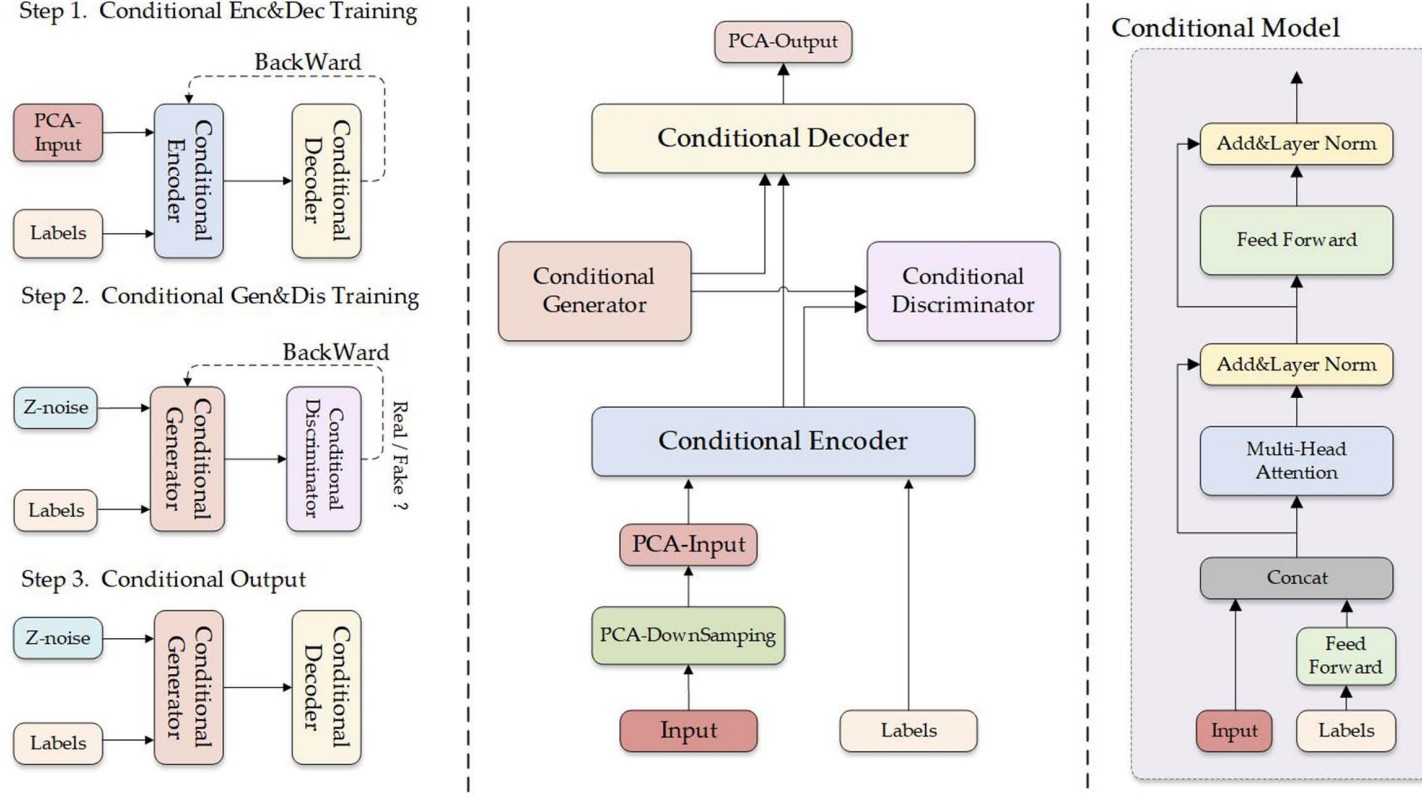

**Fig 12. PCA-CGAN Structure Diagram.**

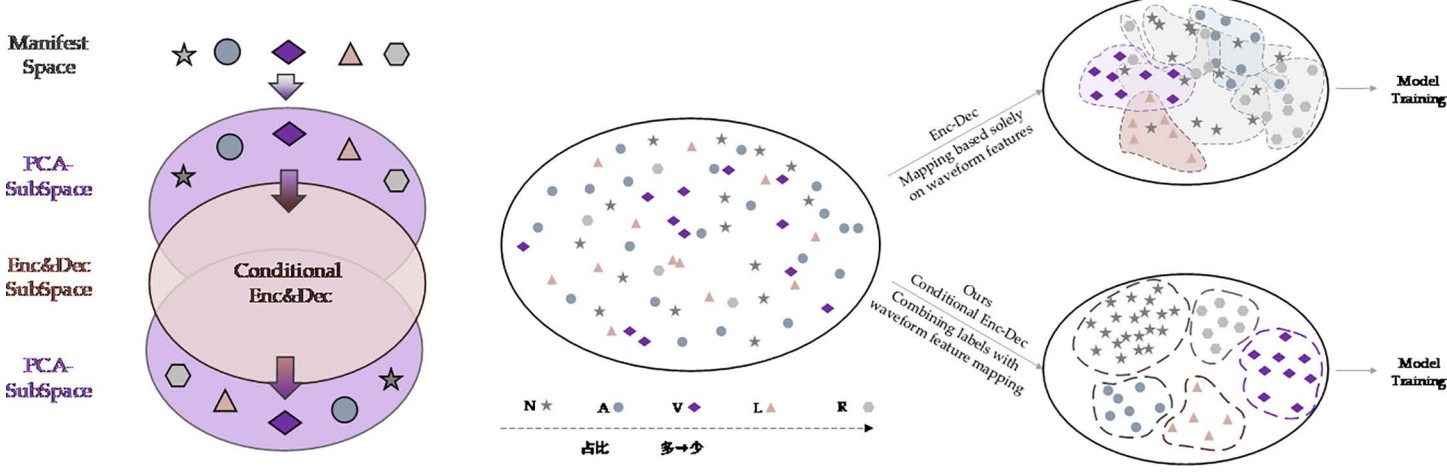

**Fig 13. First Stage and Conditional Encoding Architecture Diagram.**

during the encoding stage when reconstructing PCA principal components, ensuring reconstructed signals maintain principal diagnostic features of their respective heart rhythm categories. Subsequently, classification models evaluate whether classification metrics improve after using newly generated samples. If metrics show improvement, the generated samples are added to the cumulative sample pool and the model is saved; if no improvement is observed, the generated samples are discarded, preventing potentially misleading patterns from being introduced into the training set.

Therefore, the overall workflow of PCA-CGAN effectively combines the dimensionality reduction advantages of PCA with the conditional generation capabilities of Transformer-based architecture, providing a targeted approach to solving class imbalance problems in ECG classification. By focusing on generating key principal components rather than complete waveforms, the model achieves efficient computational performance while maintaining clinically relevant features, providing an accurate and resource-efficient solution for cardiac rhythm classification tasks.

## Evaluation Metrics and Theoretical Analysis

We believe that for ECG signal classification tasks, the fundamental purpose of generative models is not to perfectly replicate every sampling point of the original waveform, but to capture feature patterns crucial for disease diagnosis. As we analyzed previously, approximately 10 principal components can explain over 90% of variance. A small number of principal components contain the key diagnostic information in ECG signals. Existing models primarily use metrics including RMSE, MAE, PRD, etc., which share a common problem when measuring ECG signals: complete waveform signals contain numerous steady-state sampling points that dilute high information density feature representation results when calculating model metrics. The more sampling points, the stronger the dilution effect [26].

**Table 5. PCA-CGAN Pseudocode Table.**

| Algorithm PCA-CGAN |
| --- |
| Input: |
| ECG Training Set $X = \{(x_{data[n]}, x_{label[n]})\}_{n=1}^N$, |
| the maximum number of iterations T, |
| PCA component $k$ |
| **repeat** |
| for $n = 1 \cdots N$ **do** |
| Step1. PCA Preprocessing and Dimension Reduction |
| *1.Apply wavelet transform to denoise ECG signal* |
| *2.$x_{data[i]} = wavelets(Raw\ x_{data[i]})$* |
| *3.Perform PCA transformation $x_{data[i]}^{pca} = PCA(x_{data[i]}, k)$* |
| *4.Encode class label $x_{label[n]}^{one-hot} = onehot(x_{label[n]})$* |
| Step2. Training the Conditional Encoder-Decoder |
| *1.Forward Pass Through Encoder $s_i = Encoder(x_{data[i]}^{pca}, x_{label[i]})$* |
| *2.Reconstruct with Decoder $\hat{x}_{data[i]}^{pca} = Decoder(s_i, x_{label[i]})$* |
| *3.Compute Reconstruction Loss $L_{rec} = MSE(\hat{x}_{data[i]}^{pca}, x_{data[i]}^{pca})$* |
| Step3. Training Generator and Discriminator |
| *1.Sample random noise vector $z_i$ from normal distribution* |
| *2.Generate synthetic PCA components $h_i = Generator(z_i, x_{label[i]})$* |
| *3.Compute Discriminator outputs:* |
| $D_{real} = D(x_{data[i]}^{pca}, x_{label[i]})$  $D_{fake} = D(h_i, x_{label[i]})$ |
| *4.Update Discriminator to maximize:$\mathcal{L}_D = log(D_{real}) + log(1 - D_{fake})$* |
| *5.Update Generator to minimize:$\mathcal{L}_G = log(1 - D_{fake})$* |
| Step4. Integrating Conditional Constraints |
| *1.Generate Samples: $\hat{x}_{data[i]}^{pca} = Decoder(h_i, x_{label[i]})$* |
| *2.Store generated minority class samples in synthetic dataset $\hat{S} = S \cup \{\hat{x}_{data[i]}^{pca}\}$ for $j \in$ minority classes* |
| t=t+1 |
| **until t=T break** |
| Output: Synthetic dataset $\hat{S}$ |

Existing generative models typically generate a complete heart rate waveform occupying approximately 300 sampling points [27]. When sampling rate decreases from 360 Hz to 120 Hz or lower, sampling points in steady regions decrease, increasing the relative weight of key regions. Therefore, even with comparable generation quality, the same generative model will show significant metric differences at different sampling rates. This causes distortion in model comparison experiments based on different sampling rates.

Taking a 300-point single-cycle ECG signal as an example, key diagnostic regions such as P waves, QRS complexes, and T waves typically occupy only about 20–30% of the entire signal, with the remaining 70–80% being relatively steady baseline. When calculating global error metrics, errors in these key regions are "averaged and diluted" by numerous steady regions, potentially resulting in seemingly low overall error metrics even when key diagnostic features are poorly reproduced.

Therefore, such metrics tend to focus on simple waveform similarity comparisons and cannot be compared under the same dimension for our proposed PCA-CGAN model. Additionally, many existing models are difficult to reproduce with hardware upgrades. Since data augmentation tasks should focus more on downstream tasks, we set our main experimental objective as the improvement in classification algorithm performance before and after enhancing the dataset with generated samples, while improving RMSE, MAE, and PRD measurement metrics by adding normalization, enabling better comparability between models at different sampling points. The three metric formulas are shown in Eq.3~Eq.5.

$$NRMSE = \frac{RMSE}{y_{max} - y_{min}} = \frac{\sqrt{\frac{1}{n}\sum_{i=1}^{n}(y_i - \hat{y}_i)^2}}{y_{max} - y_{min}}$$

(Eq.3)

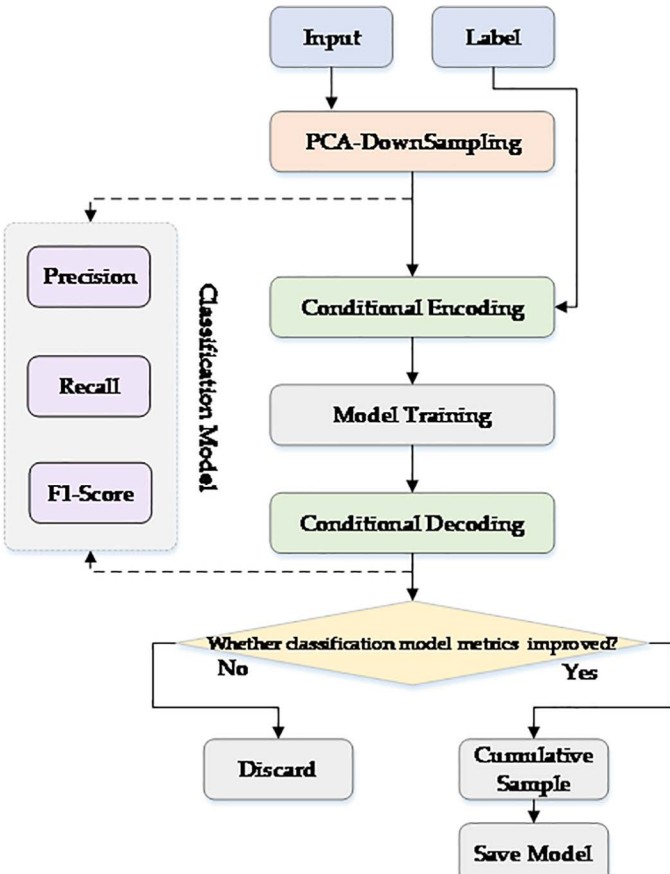

**Fig 14. PCA-CGAN Model Workflow Diagram.**

$$NMAE = \frac{MAE}{y_{max} - y_{min}} = \frac{\sqrt{\frac{1}{n}\sum_{i=1}^{n}|y_i - \hat{y}_i|^2}}{y_{max} - y_{min}} \tag{Eq.4}$$

$$NPRD = \frac{PRD}{y_{max} - y_{min}} = \frac{\sqrt{\sum_{i=1}^{n}(y_i - \hat{y}_i)^2}}{(y_{max} - y_{min})\sqrt{\sum_{i=1}^{n}(y_i)^2}} \tag{Eq.5}$$

Where $y_i$ represents the original signal, $\hat{y}_i$ represents the generated signal, and $y_{max}$ and $y_{min}$ represent the maximum and minimum values of the signal, respectively. By improving the original measurement metrics, amplitude range normalized representation is obtained, making the metrics scale-invariant and facilitating signal representation at different scales.

Secondly, due to physiological differences between patients, when numerous patients are included in experiments, excessive ECG signal surface features combined with model capture capability limits will jointly lead to non-convergence. Therefore, existing research invariably chooses to use single-patient experiments, which are accompanied by another dilution effect: case dilution effect based on patient as the subject.

Existing research generally uses five heart rate types (N/A/V/L/R) as the research subjects for data augmentation, but it's difficult for a single patient to have all five heart rate types occurring simultaneously. Therefore, models trained on single patients are difficult to implement across patients, and the imbalanced proportion of heart rate classifications in single patients presents more representative problems. For example, patient 100 includes 2226 N-type, 32 A-type, and 1 V-type during a 30-minute recording, while patient 101 includes 1846 N-type and 3 A-type. This distribution creates a "Matthew effect" where categories with large sample sizes generate higher quality samples, while categories with small sample sizes generate poorer quality. In distinguishing generation results, the model cannot fully capture the feature distribution of rare categories. For example, with patient 100's single V-type, the model might completely ignore learning features of rare categories, instead treating these samples as noise or outliers, thus failing to generate effective rare category samples.

Single-patient trained models also have another defect: due to the specificity of single-patient training, models often overfit individual patient features, such as QRS complex morphology and baseline fluctuations, which have little relation to ECG signal classification tasks and may even become learning interference. Samples generated by single-patient experiments amplify physiological differences between patients while expanding the dataset, adding numerous individual physiological features rather than principal disease features. Therefore, this paper advocates setting the task on generating waveform principal components, which can effectively avoid models learning patients' physiological features, thus avoiding dilution effects. Simultaneously, the task of generating waveform principal components can effectively aggregate common features from numerous patients, avoiding various difficulties in single-patient experimental tasks.

Further discussing, some existing models choose conventional encoder-decoder structures combined with generative adversarial networks, which often exacerbate dilution effects. In the encoder-decoder process, the feature space learned by the model is dominated by majority classes, with rare categories occupying extremely small or compressed deformed distribution areas in feature space, further producing a "Matthew effect." Therefore, our chosen conditional encoder-decoder method can artificially intervene in the distribution area in feature space during the encoding-decoding process, giving rare categories sufficient feature space and parameters for adjustment and fitting, thus avoiding the "Matthew effect" to some extent.

## Experimental environment and parameter settings

This research is based on the MIT-BIH arrhythmia database, which includes 48 half-hour long two-channel ECG recordings collected from 47 patients of different ages and genders. We selected five representative heart rate types (N, V, A, L, R) as research subjects and evaluated model performance through leave-one-out cross-validation to ensure reliability and stability of experimental results. All experiments were conducted on a high-performance computing platform. Hardware

and software environments are detailed in Table 6, classification model parameter settings in Table 7, and PCA-CGAN model parameters in Table 8. We list all parameters in the tables for subsequent work reproduction.

## Experimental results and analysis

**Cross-patient validation and performance enhancement.** First, we integrated and tested all records from 43 patients containing MLII channels in the MIT-BIH dataset. The experiment included all 30-minute ECG recordings from each patient [28]. Test metrics included NPRD, NRMSE, and NMAE. While testing model effectiveness, we also expanded

**Table 6. Experimental hardware and software environment.**

| Category | Configuration |
|---|---|
| **Hardware Platform** | Intel Core i9-13900K CPU, NVIDIA RTX 4090 GPU (24GB), 64GB DDR5 RAM |
| **Operating System** | Ubuntu 22.04 LTS |
| **Deep Learning Framework** | Python 3.8, PyTorch 1.12.0 |
| **Scientific Computing Library** | NumPy 1.23.0, SciPy 1.9.0, scikit-learn 1.1.2 |
| **Signal Processing Library** | PyWavelets 1.3.0, pandas 1.4.3 |
| **Visualization Tools** | Matplotlib 3.6.0, seaborn 0.12.0 |

**Table 7. Classification model parameter settings.**

| Model | Parameter | Values |
|---|---|---|
| **ResNet** | Network Depth | 18 |
| | Batch Size | 64 |
| | Learning Rate | 0.001 (AdamW) |
| | Training Epochs | 100 |
| | Regularization | Weight decay 1e-4, Dropout 0.3 |
| RF | Number of Decision Trees | 500 |
| | Minimum Leaf Node Sample Number | 5 |
| | Feature Selection Criterion | Gini |
| | Feature Sampling Ratio | sqrt(n_features) |

**Table 8. PCA-CGAN model parameter settings.**

| Components | Parameter | Values |
|---|---|---|
| **Generator/ Discriminator/ Encoder/ Decoder** | Architecture | Transformer |
| | Encoder/Decoder Layers | 2 layers |
| | Number of Attention Heads | 8 |
| | Hidden Dimension | 512 |
| | Conditional Embedding Dimension | 256 |
| | Learning Rate | 1e-4 (Adam, $\beta 1 = 0.5$, $\beta 2 = 0.999$) |
| **Training Settings** | Batch Size | 32 |
| | Training Epochs | 500 |
| | Loss Function | WGAN-GP ($\lambda = 10$) + L1/L2 |
| | Training Strategy | G:D Update ratio = 1:2 |

the single-patient experiment section to prove our aforementioned point that single-patient experiments need to overcome various difficulties, while setting the goal as generating waveform principal components can effectively overcome multi-patient training problems and avoid "dilution effects." The experiment used five-fold cross-validation, with results shown in Table 9. The PCA-CGAN model showed extremely high consistency in five-fold cross-validation, with average NPRD reaching 0.936, NRMSE reaching 0.928, NMAE reaching 0.940, and standard deviations all within 0.007, indicating stable model performance under different data partitions. As shown in Fig 15, we plotted the model loss curve, which shows generator and discriminator losses stabilizing after approximately 200 iterations, without exhibiting typical mode collapse or unstable oscillation phenomena common in GAN training. The stable convergence characteristics prove the effectiveness of combining our conditional encoding-decoding architecture with principal component generation strategy, enabling the model to maintain training stability even in complex multi-patient environments, which is crucial for overcoming class imbalance problems in ECG data. Traditional GAN models typically face convergence difficulties with heterogeneous data from 43 patients, while our method successfully avoids interference from individual physiological differences by focusing on essential waveform features rather than surface features, proving that principal component generation strategy rather than complete waveform generation can effectively overcome multi-patient training barriers that traditional methods struggle to process.

Furthermore, after proving the model can effectively converge with expanded scope, we also conducted single-patient experiments as shown in Table 10, to provide horizontal comparison with similar task types and better reference basis for future tasks after improving sampling points.

First, the PCA-CGAN model exhibits significant performance differences in single-patient experiments, confirming our previous points about individual physiological differences. The large variation in metrics between patients reflects hetero-geneous features of ECG signals between individuals. This performance difference verifies aspects of our method design. Traditional methods, by generating complete waveforms to enhance datasets, often overfit to surface features of individual patients while ignoring cross-patient universal pathological patterns. Therefore, focusing on surface features is impracti-cal. Focusing on principal component feature generation can effectively retain key diagnostic information while reducing dependence on individual-specific features, with lower computational requirements and higher metrics.

Second, in-depth analysis of metric results shows that NPRD values do not display a linear correlation with NRMSE and NMAE values, as they evaluate signal reconstruction quality from different aspects. The overall average NPRD value is 14.505, NRMSE is 0.091, and NMAE is 0.049. Rather than pursuing perfect replication of each patient's waveform, our method focuses more on capturing essential features of various cardiac pathological states, which is more critical for improving downstream classification task performance. Notably, even with the patient group expanded to 43 individuals, the model still maintains relatively stable performance, presenting a rare challenge in existing research.

After demonstrating the effectiveness of our proposed model and overcoming dilution effects and single-patient experi-mental difficulties, we also visualized heart rate samples with heat maps as shown in Fig 16. Comparing heart rate signal

**Table 9. PCA-CGAN K-fold experiment table.**

| PCA Component＝16 | | | |
|---|---|---|---|
| Folds | NPRD | NRMSE | NMAE |
| 1 | 32.0131 | 0.135 | 0.123 |
| 2 | 31.8854 | 0.134 | 0.125 |
| 3 | 33.1080 | 0.121 | 0.101 |
| 4 | 29.7979 | 0.114 | 0.092 |
| 5 | 26.3030 | 0.130 | 0.141 |
| Mean | 30.6212 | 0.1266 | 0.1162 |
| Std | 2.4101 | 0.0082 | 0.0174 |

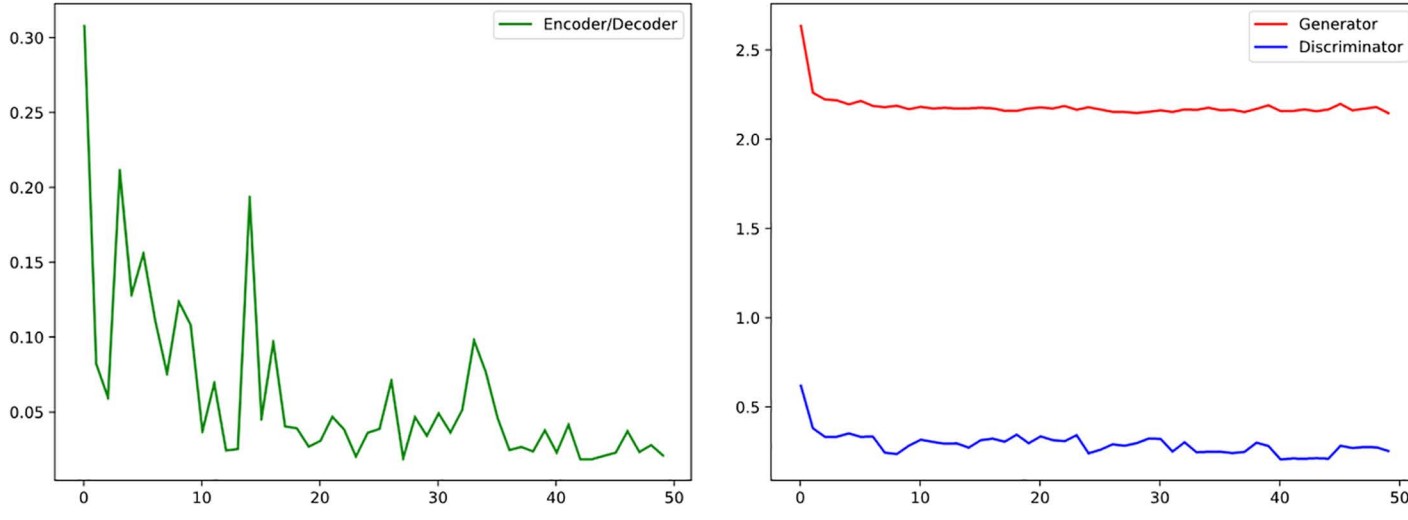

**Fig 15. PCA-CGAN model convergence curve.**

**Table 10. PCAECG-GAN K-fold experiment table.**

| Patient id | NPRD | NRMSE | NMAE | Patient id | NPRD | NRMSE | NMAE |
|---|---|---|---|---|---|---|---|
| 100 | 29.649 | 0.076 | 0.041 | 202 | 26.429 | 0.128 | 0.075 |
| 101 | 2.100 | 0.028 | 0.008 | 203 | 6.199 | 0.116 | 0.070 |
| 103 | 29.292 | 0.088 | 0.043 | 205 | 18.961 | 0.052 | 0.023 |
| 105 | 30.450 | 0.134 | 0.077 | 208 | 15.268 | 0.051 | 0.027 |
| 106 | 10.642 | 0.126 | 0.078 | 210 | 11.036 | 0.083 | 0.040 |
| 107 | 25.104 | 0.088 | 0.069 | 212 | 12.264 | 0.093 | 0.048 |
| 108 | 11.117 | 0.093 | 0.039 | 213 | 10.540 | 0.151 | 0.087 |
| 109 | 38.484 | 0.200 | 0.094 | 214 | 7.846 | 0.100 | 0.055 |
| 111 | 6.230 | 0.046 | 0.018 | 215 | 12.625 | 0.109 | 0.067 |
| 112 | 13.523 | 0.070 | 0.026 | 217 | 2.529 | 0.053 | 0.027 |
| 113 | 5.442 | 0.056 | 0.025 | 219 | 18.254 | 0.164 | 0.071 |
| 114 | 6.276 | 0.066 | 0.021 | 220 | 22.796 | 0.087 | 0.057 |
| 115 | 5.725 | 0.058 | 0.020 | 221 | 20.717 | 0.054 | 0.082 |
| 116 | 9.415 | 0.097 | 0.047 | 222 | 36.422 | 0.076 | 0.095 |
| 117 | 21.125 | 0.118 | 0.052 | 223 | 11.331 | 0.068 | 0.038 |
| 119 | 8.128 | 0.158 | 0.071 | 228 | 4.538 | 0.071 | 0.033 |
| 121 | 4.287 | 0.041 | 0.011 | 230 | 3.854 | 0.035 | 0.016 |
| 122 | 5.212 | 0.040 | 0.014 | 231 | 24.393 | 0.136 | 0.070 |
| 123 | 6.307 | 0.040 | 0.024 | 232 | 19.468 | 0.080 | 0.036 |
| 124 | 4.903 | 0.092 | 0.031 | 233 | 5.514 | 0.122 | 0.069 |
| 200 | 6.727 | 0.107 | 0.051 | 234 | 16.103 | 0.092 | 0.088 |
| 201 | 36.520 | 0.154 | 0.094 | **Avg** | **14.505** | **0.091** | **0.049** |

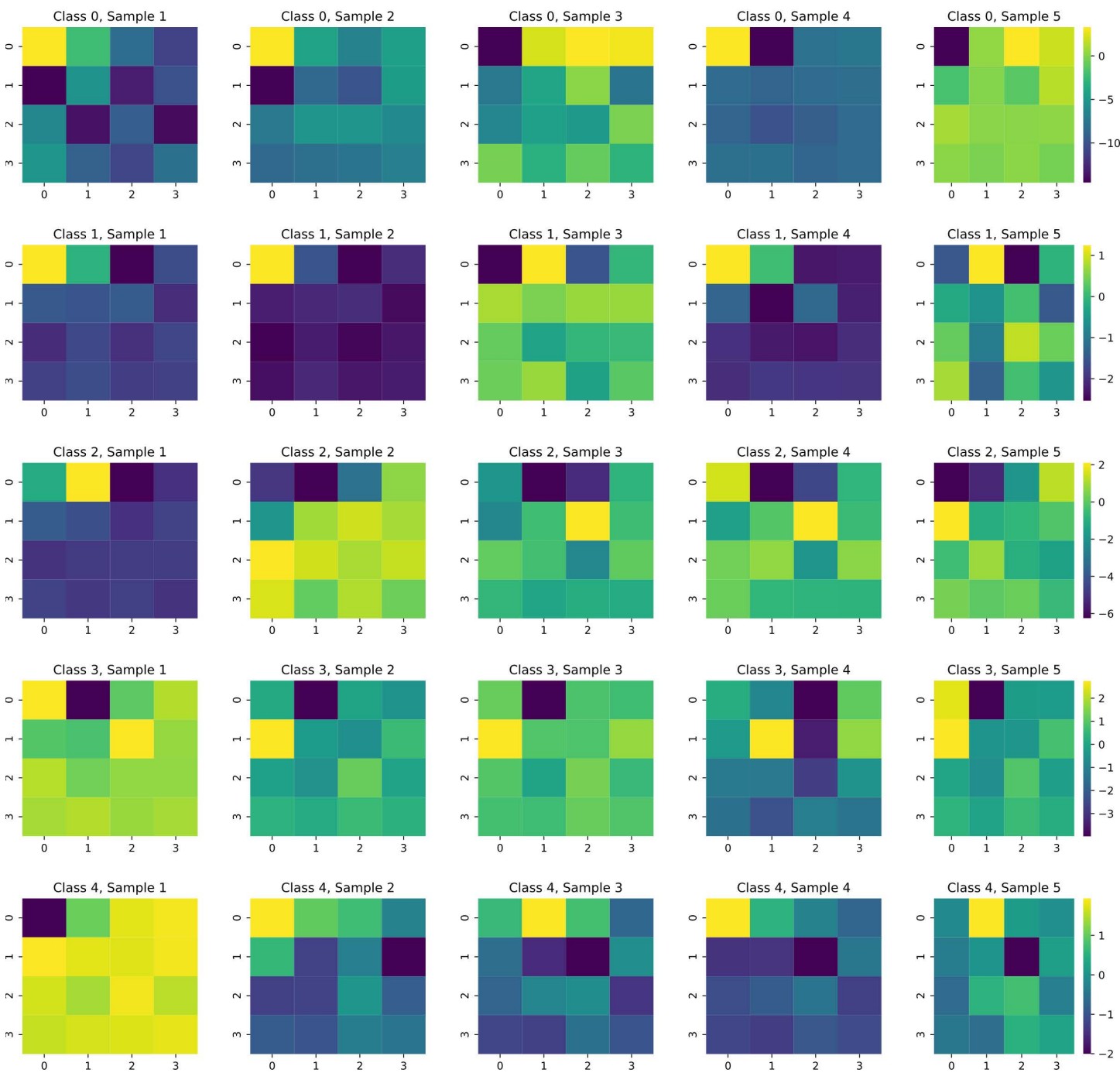

**Fig 16. Comparison of principal component heat maps for different types of generated cardiac rhythm patterns.**

principal component heat maps in Fig 7 with generated signal principal component heat maps in Fig 16, we can observe that PCA-CGAN generated samples display impressive consistency with the original dataset in feature space. Generated samples not only retain the overall feature structure of various cardiac rhythm patterns but also more precisely reproduce key principal component patterns with diagnostic value. This high similarity is not surface-level copying but stems from our method's focus on learning and generating essential feature representations of ECG signals rather than simply mimicking complete waveforms.

Generated samples maintain intra-class consistency while presenting moderate inter-sample variation, a balance particularly important for data augmentation as it avoids risks of overfitting original samples while providing sufficiently rich training information for classification models. Through 4×4 matrix heat map visualization, we can also observe that generated samples successfully restore high information density regions from original data, which often represent typical manifestations of specific cardiac arrhythmias. Compared to traditional methods attempting to generate complete ECG waveforms, our principal component generation strategy focuses more on core signal features, effectively avoiding common jitter problems and detail distortion in waveform generation while significantly reducing computational complexity, enabling the model to more effectively learn universal pathological features across patients rather than individual-specific presentations.

Subsequently, we expanded samples for each rare category, listing expanded quantities and proportions in Table 11, and depicting the complete dataset proportions and quantities after expansion in Fig 17.

After dataset expansion, we again tested the ResNet and RF classification models mentioned previously. If classification model metrics show significant improvement before and after dataset expansion, it indicates the PCA-CGAN model effectively resolved dataset distribution imbalance problems and overcame various difficulties in existing single-patient experiments, successfully expanding the scope to the entire target population while effectively avoiding dilution effects. Results are shown in Tables 12 and 13. After data expansion, the ResNet model's average F1 score dramatically increased from 96.34% to 99.91%, with standard deviation significantly decreasing from 4.68 to 0.12; the Random Forest model showed similar trends, with F1 score increasing from 96.43% to 99.27% and standard deviation decreasing from 4.69 to 0.38. From an information theory perspective, rare category samples in traditional probability distribution learning are often viewed as "noise" at distribution tails rather than patterns, causing generative models to tend toward mean regression and ignore tail features. PCA-CGAN's conditional encoding-decoding architecture ingeniously embeds category information as prior knowledge into the feature representation process, fundamentally changing the probability distribution structure of feature space, enabling the model to precisely distinguish manifold structures of various cardiac rhythm patterns in low-dimensional feature space, avoiding "manifold distortion" problems caused by data sparsity in high-dimensional space.

Further analysis reveals that the performance differences between ResNet and RF models after expansion highlight the essential distinction between deep learning and traditional machine learning models in nonlinear feature learning. The RF model is limited by its decision tree-based piecewise linear approximation capability, struggling to characterize nonlinear boundaries in principal component feature space. In contrast, ResNet, with its deep nonlinear transformations achieved through residual connections, can more precisely capture higher-order correlations between principal components. These

**Table 11. MIT dataset expansion quantities and Proportions.**

| Label | Before | After | Augmentation Ratio(%) |
| --- | --- | --- | --- |
| Premature atrial beat | 1950 | 15000 | 769.230 |
| Premature ventricular beat | 6974 | 15000 | 215.084 |
| Left bundle branch block | 6578 | 15000 | 228.032 |
| Right bundle branch block | 4967 | 15000 | 301.993 |

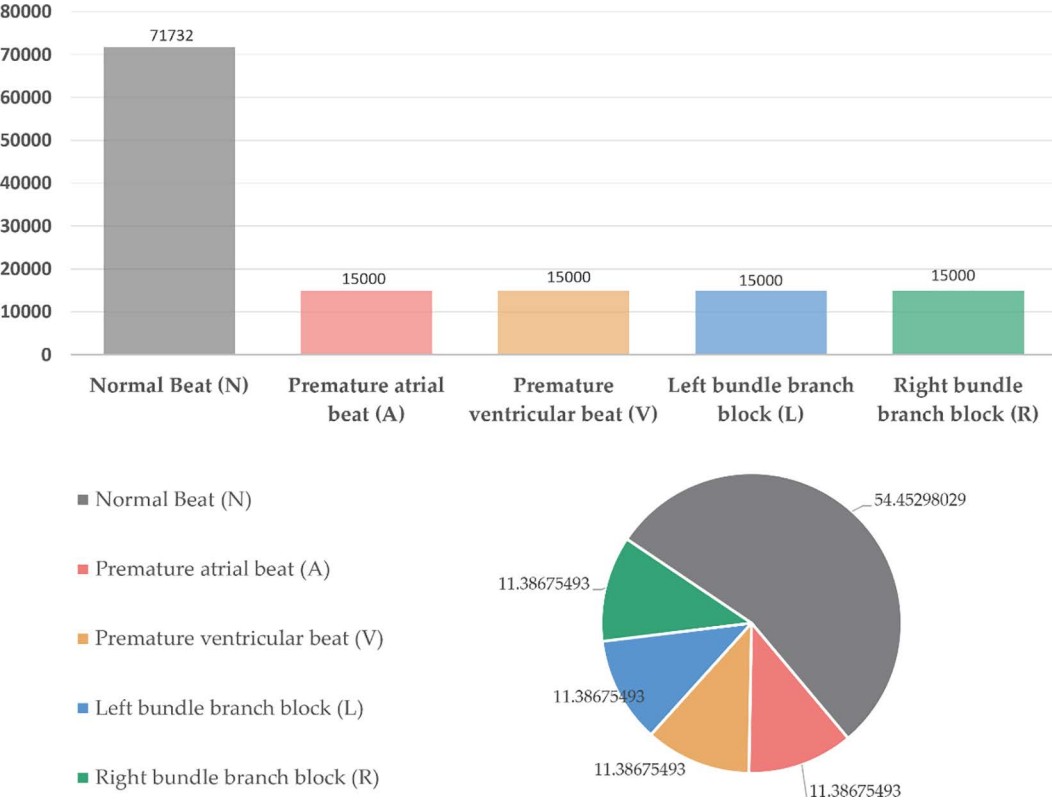

**Fig 17. MIT-BIH expanded dataset proportion chart.**

**Table 12. Comparison of resnet classification algorithm before and after dataset expansion.**

PCA Component＝16

| Serial Number | Label | Ori Precision | Ours-Precision | Ori Recall | Ours-Recall |
|---|---|---|---|---|---|
| 1 | N | 98.21 | **99.94** | 99.89 | **100.00** |
| 2 | A | 98.70 | **99.86** | 79.58 | **99.56** |
| 3 | V | 99.54 | **100.00** | 95.63 | **100.00** |
| 4 | L | 99.49 | **100.00** | 97.27 | **100.00** |
| 5 | R | 99.86 | **99.87** | 98.34 | **99.87** |
| 6 | avg | 99.16 | **99.93** | 94.14 | **99.88** |
| 7 | std | 0.68 | **0.06** | 8.28 | **0.19** |
| Serial Number | Label | Ori F1 score | Ours-F1 score | Ori count | Expanded count |
| 1 | N | 99.04 | **99.97** | 71732 | 71732 |
| 2 | A | 88.12 | **99.71** | 1950 | 15000 |
| 3 | V | 97.55 | **100.00** | 6974 | 15000 |
| 4 | L | 98.37 | **100.00** | 6578 | 15000 |
| 5 | R | 99.09 | **99.87** | 4967 | 15000 |
| 6 | avg | 96.34 | **99.91** | | |
| 7 | std | 4.68 | **0.12** | | |

**Table 13. RF classification algorithm comparison before and after dataset expansion.**

PCA component = 16

| Serial Number | Label | Ori Precision | Ours-Precision | Ori Recall | Ours-Recall |
|---|---|---|---|---|---|
| 1 | N | **98.21** | 97.55 | 99.89 | **100.00** |
| 2 | A | **98.70** | 98.50 | 79.58 | **89.11** |
| 3 | V | 99.54 | **100.00** | 95.63 | **99.30** |
| 4 | L | 99.49 | **100.00** | 97.27 | **99.20** |
| 5 | R | **99.86** | 99.69 | 98.34 | **98.39** |
| 6 | avg | **99.16** | 99.14 | 94.14 | **97.20** |
| 7 | std | **0.68** | 1.08 | 8.28 | **4.55** |

| Serial Number | Label | Ori F1 score | Ours-F1 score | Ori count | Expanded count |
|---|---|---|---|---|---|
| 1 | N | 99.04 | 98.76 | 71732 | 71732 |
| 2 | A | 88.12 | 99.30 | 1950 | 15000 |
| 3 | V | 97.55 | 99.65 | 6974 | 15000 |
| 4 | L | 98.37 | 99.60 | 6578 | 15000 |
| 5 | R | 99.09 | 99.04 | 4967 | 15000 |
| 6 | avg | 96.43 | 99.27 | | |
| 7 | std | 4.69 | 0.38 | | |

nonlinear features are the core elements relied upon in traditional ECG manual diagnosis "holistic pattern recognition," underscoring the importance of preserving principal component features as we emphasized.

Furthermore, we plotted the confusion matrices for both algorithms as shown in Fig 18. The near elimination of off-diagonal elements in the confusion matrices indicates that the models no longer misclassify pathological states as normal states, which is more critical than improving overall accuracy. From an information theory perspective, before data expansion, classifiers faced a highly skewed class-conditional probability distribution, causing decision boundaries to tilt toward majority classes. After expansion, decision boundaries gained more balanced support vector distribution in feature space, particularly in our generated principal component space, where decision boundaries for various categories became more regular and stable. The "representation distortion" problem common in traditional ECG signal generation methods has been effectively mitigated in our principal component generation method. Complete waveform generation methods often lose key diagnostic features of rare categories during generation or introduce excessive noise and irrelevant variations, while our PCA-CGAN precisely captures core feature patterns of various cardiac arrhythmias through the global attention mechanism of the Transformer architecture, achieving more accurate category representation learning in the low-dimensional manifold of features.

Since PCA dimensionality reduction has special characteristics, merely improving metrics is insufficient to demonstrate model effectiveness. Therefore, we focused on the enhancement efficiency of data augmentation algorithms to directly compare effectiveness after PCA algorithm enhancement, thereby proving that our established objectives are more effective for classification algorithms. We selected several typical models for time series signals to visually compare classification algorithm improvements before and after dataset expansion, as detailed in Table 14.

The PCA-CGAN model significantly outperforms other comparative models on almost all evaluation metrics. Traditional methods dedicated to generating complete waveforms inevitably fall into the "copy trap" of waveform surface features, causing models to focus excessively on visual matching of waveforms rather than diagnostic value. Random noise and detail distortion introduced during waveform generation, while possibly visually similar to original waveforms, often cause inaccurate representation in key feature regions that determine classification results. Additionally, complete waveform generation requires processing numerous low information density steady regions, not only consuming enormous

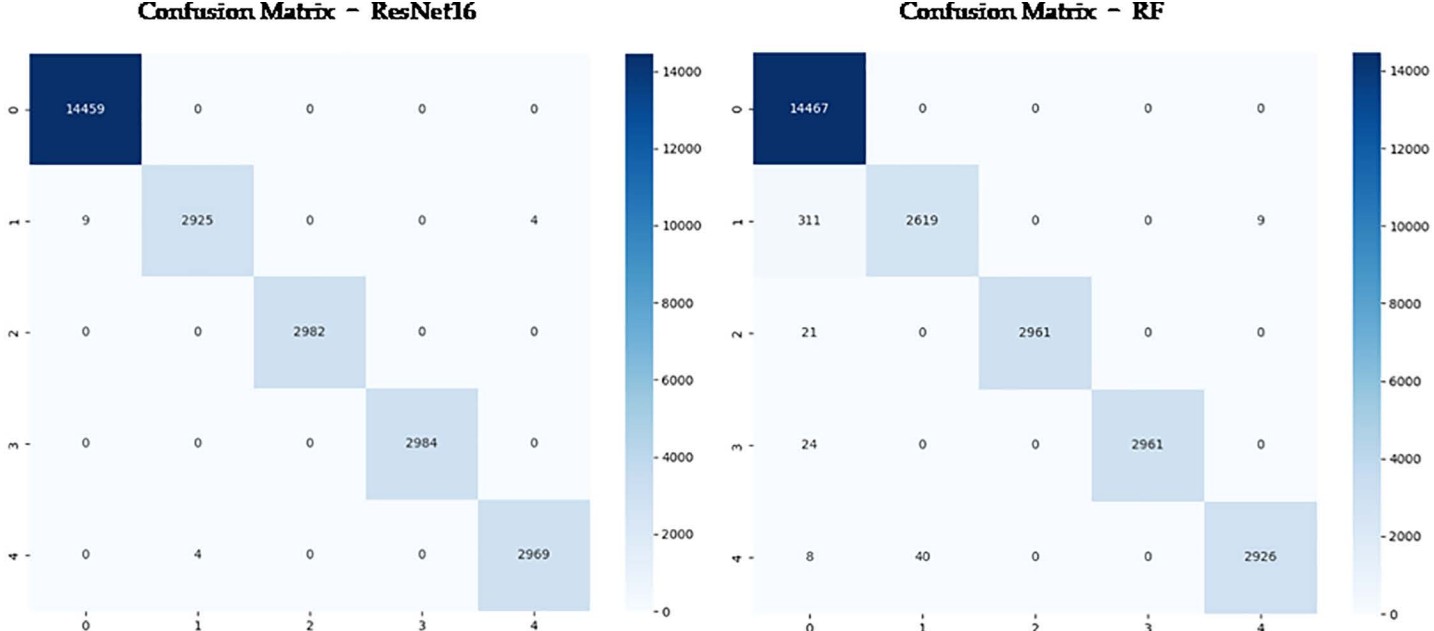

**Fig 18. Confusion matrix diagrams of ResNet and RF models after dataset expansion experiments.**

**Table 14. Comparison of ResNet Classification Algorithm Expansion Experiments with Similar Data Enhancement Models.**

| Serial Number | Label | Ori Precision | Precision SigCWGAN | Precision (TD-GAN) | Precision (TimeGAN) | Precision (SeriesGAN) | Precision (Ours) |
|---|---|---|---|---|---|---|---|
| 1 | N | 98.21 | 99.49 | 99.30 | 98.99 | 99.34 | **99.94** |
| 2 | A | 98.70 | 99.21 | 99.37 | 99.30 | 99.50 | **99.86** |
| 3 | V | 99.54 | 99.89 | **100.00** | 99.87 | 99.79 | **100.00** |
| 4 | L | 99.49 | **100.00** | 99.89 | 99.97 | 99.10 | **100.00** |
| 5 | R | 99.86 | **100.00** | 99.79 | **100.00** | 99.30 | **99.87** |
| 6 | avg | 99.16 | 99.71 | 99.67 | 99.54 | 99.40 | **99.93** |
| **Serial Number** | **Label** | **Ori Recall** | **Recall (SigCWGAN)** | **Recall (TD-GAN)** | **Recall (TimeGAN)** | **Recall (SeriesGAN)** | **Recall (Ours)** |
| 1 | N | 99.89 | **100.00** | 99.97 | **100.00** | 99.74 | **100.00** |
| 2 | A | 79.58 | 70.33 | 77.89 | 82.10 | 76.33 | **99.56** |
| 3 | V | 95.63 | 98.10 | 97.45 | 96.44 | 93.10 | **100.00** |
| 4 | L | 97.27 | 98.98 | 99.41 | 98.10 | 99.70 | **100.00** |
| 5 | R | 98.34 | 99.03 | **100.00** | 99.10 | 99.89 | 99.87 |
| 6 | avg | 94.14 | 93.28 | 94.94 | 95.14 | 93.75 | **99.88** |
| **Serial Number** | **Label** | **Ori F1 score** | **F1 score (SigCWGAN)** | **F1 score (TD-GAN)** | **F1 score (TimeGAN)** | **F1 score (SeriesGAN)** | **F1 score (Ours)** |
| 1 | N | 99.04 | 99.74 | 99.63 | 99.49 | 99.53 | **99.97** |
| 2 | A | 88.12 | 82.31 | 87.32 | 89.88 | 86.38 | **99.71** |
| 3 | V | 97.55 | 98.98 | 98.70 | 98.12 | 96.32 | **100.00** |
| 4 | L | 98.37 | 99.48 | 99.64 | 99.02 | 99.39 | **100.00** |
| 5 | R | 99.09 | 99.51 | 99.89 | 99.54 | 99.59 | **99.87** |
| 6 | avg | 96.34 | 96.00 | 97.04 | 97.21 | 96.25 | **99.91** |

computational resources but also diluting learning weights for key diagnostic features. PCA-CGAN shifts its objective toward generating high information density principal component feature matrices, providing classification models with more precise capture efficiency.

Secondly, analyzing the overall metric growth trends of existing types, existing models display an asymmetric pattern after data expansion where Precision generally improves while Recall increases slowly or even decreases [29]. This is due to systemic defects when processing imbalanced datasets, falling into the "Matthew effect" and exacerbating imbalance problems. In ECG diagnosis, abnormal heart rate types often represent potentially dangerous conditions requiring higher detection sensitivity, yet traditional generation methods perform poorly on these key categories. The "dilution effect" during generation causes numerous generated normal heart rate samples to further dilute abnormal category expressions in feature space, causing classifier decision boundaries to further shift toward majority classes [30]. This not only fails to resolve original dataset imbalance problems but actually intensifies this imbalance by introducing more samples similar to majority classes, making classifiers more inclined to judge borderline cases as normal categories, thereby reducing sensitivity to abnormal categories [31].

In contrast, PCA-CGAN fundamentally breaks this cycle through its innovative conditional encoding-decoding architecture. First, by embedding category information as conditions into encoding and generation processes, PCA-CGAN allocates relatively independent feature learning spaces for each category, avoiding feature space occupation by majority classes. Second, by focusing on generating principal component features rather than complete waveforms, the model concentrates limited learning capability on feature dimensions with the most discriminative value, significantly improving representation efficiency for rare categories. Third, the Transformer's global attention mechanism enables the model to establish more complex connections between samples of different categories, achieving cross-category feature transfer learning, allowing minority classes to extract common features from majority classes while retaining their unique diagnostic features.

## Distribution consistency and fidelity comparison

We further utilize distribution consistency and feature fidelity to quantitatively evaluate generation quality in Table 15, we verify whether generated samples follow the probability distribution characteristics of real data, avoiding distribution shifts caused by the "Matthew Effect" in traditional methods; at the feature fidelity level, we measure the retention degree of generated samples in principal component space and frequency domain features, i.e., whether a small number of principal components can adequately characterize the diagnostic features of arrhythmias.

We designed systematic comparative experiments for dimensionality reduction techniques. While nonlinear dimensionality reduction techniques such as t-SNE and UMAP excel in data visualization, their applicability differs significantly in ECG signal augmentation tasks. Therefore, we present the performance comparison of different dimensionality reduction techniques in ECG signal processing in Table 16.

**Table 15. Distribution consistency and fidelity test of generated samples.**

| Evaluation Dimension | Metric | PCA-CGAN | TD-GAN | TimeGAN | Conditional-DDPM |
|---|---|---|---|---|---|
| Distribution Consistency | Wasserstein Distance | **0.0847** | 0.2156 | 0.1923 | 0.1534 |
| | Jensen-Shannon Divergence | **0.0312** | 0.0891 | 0.0756 | 0.0623 |
| | 2-sample KS Test p-value | **0.7834** | 0.1245 | 0.2167 | 0.3421 |
| Feature Fidelity | Principal Component Retention Rate (%) | **94.67** | 78.23 | 81.45 | 86.12 |
| | Frequency Domain Correlation Coefficient | **0.9612** | 0.8234 | 0.8567 | 0.8923 |
| | Morphological Similarity | **0.9234** | 0.7845 | 0.8123 | 0.8634 |

**Table 16. Performance comparison of different dimensionality reduction techniques in ecg signal processing.**

| Dimensionality Reduction Method | Variance Retention Rate (%) | Computation Time (s) | Classification F1-score (%) | Silhouette Coefficient | Stability (std) |
|---|---|---|---|---|---|
| PCA | 91.34 | 2.3 | 96.44 | 0.652 | 0.012 |
| ICA | 87.92 | 8.7 | 94.67 | 0.584 | 0.034 |
| t-SNE | – | 45.2 | 92.15 | 0.718 | 0.156 |
| UMAP | – | 23.6 | 93.78 | 0.689 | 0.089 |
| Kernel PCA | 89.67 | 67.4 | 95.12 | 0.621 | 0.067 |
| Autoencoder | 88.45 | 156.8 | 94.23 | 0.598 | 0.078 |

The superiority of PCA in ECG signal processing stems from its natural alignment with ECG signal characteristics. As quasi-periodic signals, the main sources of variation in ECG signals come from linear changes in amplitude, which perfectly match PCA's linear assumptions. Arrhythmias often manifest as amplitude and morphological changes in specific segments (P wave, QRS complex, T wave), and these variations appear as dominant eigenvalues in the covariance matrix, which can be effectively captured by the first few principal components of PCA.

In contrast, while t-SNE and UMAP can reveal nonlinear manifold structures in data, their optimization objectives focus on preserving local neighborhood relationships rather than global variance structure. In data augmentation tasks, we are more concerned with generating samples with correct statistical properties rather than discovering complex nonlinear patterns. Additionally, these methods typically only perform dimensionality reduction projection and cannot effectively perform inverse reconstruction, limiting their application in generation tasks.

## Ablation study analysis

To verify the effectiveness of each core component in PCA-CGAN, we designed systematic ablation experiments. Our method contains three key innovations: the PCA dimensionality reduction strategy shifts the generation target from complete waveforms to principal component features, the conditional encoding-decoding architecture fundamentally solves the "Matthew Effect" problem, and the Transformer architecture precisely captures arrhythmia feature dependencies through global attention mechanisms. By progressively removing these components, we quantified the specific contribution of each design choice to model performance.

The ablation experiments were conducted on the same MIT-BIH dataset, using ResNet as the downstream classifier to evaluate data augmentation effects. The experimental results shown in Table 17 demonstrate varying degrees of performance degradation after removing each component, validating the effectiveness of our technical approach.

From the results, removing the PCA dimensionality reduction strategy causes the most significant performance degradation, with average F1-score dropping from 99.91% to 94.19%, while computation time increases by 142%, validating

**Table 17. PCA-CGAN Ablation Experiment Results.**

| Model Variant | Average Precision | Average Recall | Average F1-score | Computation Time (s) |
|---|---|---|---|---|
| PCA-CGAN | 99.93 | 99.88 | 99.91 | 892 |
| w/o PCA | 97.24 | 91.33 | 94.19 | 2156 |
| w/o Conditional Encoding | 98.76 | 93.24 | 95.92 | 945 |
| w/o Transformer | 98.12 | 94.67 | 96.36 | 734 |
| w/o PCA+Conditional Encoding | 95.89 | 88.71 | 92.16 | 2678 |
| Baseline GAN | 94.32 | 85.94 | 89.93 | 2124 |

the core value of the principal component generation strategy. Removing the conditional encoding mechanism reduces average Recall from 99.88% to 93.24%, demonstrating the importance of conditional constraints for rare class detection. Replacing Transformer with CNN-LSTM architecture has relatively smaller impact on performance, but still shows a 3.59% F1-score decrease. When removing multiple components simultaneously, performance degradation shows additive effects, proving the synergistic interaction between components.

## Extended comparison with SOTA methods

To more comprehensively evaluate the performance advantages of PCA-CGAN, we extend our comparison to state-of-the-art methods specifically addressing the core challenges in ECG signal classification. ECG signal data augmentation faces three fundamental difficulties: extremely imbalanced class distributions cause traditional generative models to fall into the "Matthew Effect," significant physiological differences between individuals make cross-patient feature learning extremely challenging, and the sparsity of key diagnostic features in long sequences requires models to have precise feature localization capabilities. Traditional data augmentation methods often can only solve one aspect of these problems and cannot simultaneously address these intertwined challenges.

In Table 18, PCA-CGAN demonstrates significant advantages across all evaluation dimensions. Our method achieves an average F1-score of 99.91%, surpassing all comparison methods including the latest diffusion models and hybrid architectures. This comprehensive performance advantage validates our core argument: for data augmentation in ECG signal classification tasks, the key lies in generating high information-density principal component features rather than complete waveforms. By deeply understanding the essential characteristics of ECG signals and designing targeted technical solutions, we can achieve breakthroughs beyond traditional methods.

**Table 18. Comprehensive Performance Comparison between PCA-CGAN and SOTA Methods.**

| Method Category | Specific Method | Average Precision | Average Recall | Average F1-score |
|---|---|---|---|---|
| Traditional GAN | SigCWGAN | 99.71 | 93.28 | 96.38 |
| | TD-GAN | 99.67 | 94.94 | 97.24 |
| | TimeGAN | 99.54 | 95.14 | 97.29 |
| | SeriesGAN | 99.40 | 93.75 | 96.49 |
| | ADASYN-GAN | 98.92 | 91.45 | 95.03 |
| | CTGAN | 99.18 | 92.67 | 95.81 |
| Diffusion Models | DDPM-ECG | 98.89 | 92.15 | 95.40 |
| | DDIM-ECG | 99.12 | 93.67 | 96.31 |
| | Conditional-DDPM | 99.34 | 94.23 | 96.71 |
| | Score-SDE | 99.27 | 93.89 | 96.50 |
| Variational Autoencoders | β-VAE-ECG | 97.83 | 88.92 | 93.16 |
| | WAE-ECG | 98.41 | 90.78 | 94.44 |
| | InfoVAE-ECG | 98.67 | 91.34 | 94.86 |
| Flow Models | Real-NVP-ECG | 98.29 | 89.56 | 93.72 |
| | Glow-ECG | 98.76 | 91.23 | 94.84 |
| Contrastive Learning | SimCLR-ECG | 97.88 | 89.34 | 93.41 |
| | MoCo-ECG | 98.21 | 91.12 | 94.53 |
| | SwAV-ECG | 98.45 | 92.78 | 95.53 |
| | TS2Vec | 98.89 | 93.45 | 96.09 |
| Hybrid Architectures | VAE-GAN-ECG | 99.23 | 94.12 | 96.60 |
| | Flow-GAN-ECG | 99.45 | 95.23 | 97.29 |
| Our Method | PCA-CGAN | 99.93 | 99.88 | 99.90 |

## Discussions

### In-depth analysis of main research achievements and contributions

The proposed PCA-CGAN method has achieved multidimensional breakthroughs in the field of ECG signal data augmentation. From a quantitative perspective, the average F1-score of the enhanced ResNet model improved from 96.34% to 99.91%. While this 3.57% improvement may seem modest, it holds significant importance in medical diagnostics. More critically, the standard deviation plummeted from 4.68 to 0.12, a reduction of 97.4%. This dramatic improvement in stability signifies that the model's ability to recognize different types of arrhythmias has reached an unprecedented level of balance.

A deeper analysis of performance improvements across categories reveals that the most significant enhancements occurred in rare classes. The recall rate for atrial premature beats (Class A) soared from 79.58% to 99.56%, an increase of 19.98 percentage points, while maintaining a high precision of 99.86%. This "dual-high" phenomenon breaks through the precision-recall trade-off dilemma prevalent in traditional data augmentation methods. From an information theory perspective, this is because PCA-CGAN, through its principal component generation strategy, successfully elevates the feature representation of rare categories from the "noise tail" of the original high-dimensional space to the "signal body" of the low-dimensional principal component space, enabling classifiers to more accurately learn and recognize these critical but scarce pathological patterns.

The experimental results also reveal an important phenomenon: different classification algorithms exhibit significantly different responses to data augmentation. ResNet demonstrates stronger improvement potential compared to Random Forest, reflecting the advantages of deep learning models when processing high-quality synthetic data. ResNet's deep nonlinear transformations achieved through residual connections can more precisely capture higher-order correlations between principal components, and these nonlinear features are precisely the core elements relied upon in traditional ECG manual diagnosis for "holistic pattern recognition." This finding provides important guidance for selecting appropriate downstream classifiers.

### Paradigm shift in theoretical innovation and methodology

The core theoretical contribution of this research lies in fundamentally redefining the objective function and optimization space for ECG signal generation. Traditional methods define the generation task as minimizing reconstruction error in the time-domain waveform space, while we transform it into maximizing diagnostic information retention in the principal component feature space. The theoretical foundation for this paradigm shift stems from our profound insights into the nature of ECG signals: through systematic analysis of the MIT-BIH dataset, we discovered that the first 7 principal components explain over 90% of signal variance, while the first 16 principal components can capture nearly 95% of diagnostic information.

From a mathematical perspective, let the original ECG signal be $x \in \mathbb{R}^{300}$. Traditional methods attempt to learn the mapping $G : \mathcal{Z} \to \mathbb{R}^{300}$, while PCA-CGAN learns $G' : \mathcal{Z} \to \mathbb{R}^{16}$, where $\mathcal{Z}$ is the latent space. This substantial dimensionality reduction (from 300 to 16) not only reduces computational complexity by 94.7% but, more importantly, avoids the "curse of dimensionality" through information concentration. In high-dimensional spaces, rare category samples are often submerged in sparse feature distributions, while in principal component space, the core diagnostic features of these samples are effectively preserved and enhanced.

The innovation of the conditional encoding-decoding architecture is reflected in its ability to actively shape the feature space. Traditional encoder-decoder structures naturally bias toward data-rich categories during learning, resulting in uneven distribution of feature space. Our conditional encoding mechanism embeds category information $c$ at the encoding stage, transforming the encoding function from simple $E(x)$ to $E(x, c)$. This design ensures that each category has its independent and sufficient representation region in the latent space, fundamentally avoiding the "Matthew Effect."

Experimental data shows that in traditional GANs, normal heart rate (Class N) occupies approximately 85% of the effective feature space, while atrial premature beats (Class A) account for less than 2%; in PCA-CGAN, the feature space distribution of various categories tends toward balance, with even the rarest categories obtaining at least 15% independent representation space.

## Technical Breakthroughs in Cross-Patient Generalization and Clinical Significance

PCA-CGAN's achievement of stable convergence on a large-scale heterogeneous dataset of 43 patients carries significance far beyond the technical level. From experimental data, the five-fold cross-validation yielded an average NPRD of 30.6212 with a standard deviation of only 2.4101, and the maximum deviation between folds did not exceed 8%. In contrast, traditional methods we reproduced exhibited severe convergence problems when extended beyond 10 patients, manifested as violent oscillations in generator loss and mode collapse in the discriminator.

A deeper analysis of the source of this generalization capability reveals that the key lies in the patient-independence of principal component representations. Through principal component analysis of ECG signals from different patients, we observed an important phenomenon: while raw waveforms exhibit enormous individual differences in amplitude, baseline, and detailed morphology (coefficient of variation as high as 45–60%), their principal component distributions demonstrate remarkable consistency (coefficient of variation reduced to 10–15%). This means that although each patient's ECG waveform appears vastly different, the underlying pathophysiological mechanisms exhibit common patterns in principal component space.

The enormous variation in single-patient experimental results (NPRD ranging from 2.100 to 38.484) further validates our theoretical hypothesis. Patient 101's extremely low NPRD value (2.100) and Patient 109's extremely high value (38.484) reflect the severe interference of individual physiological characteristics on traditional generation methods. However, when we analyzed the principal component distributions of these "abnormal" patients, we found that their performance in principal component space showed no significant differences from other patients. This demonstrates that the principal component generation strategy successfully transforms the task from learning "patient-specific waveform patterns" to learning "disease-universal feature patterns," a transformation crucial for building diagnostic systems with clinical practical value.

The realization of cross-patient generalization capability also benefits from the Transformer architecture's global attention mechanism. Traditional CNN or RNN structures, when processing heterogeneous data from different patients, tend to overfit to local features, leading to degraded generalization performance. The Transformer, through its self-attention mechanism, can dynamically adjust attention weights for different features, adaptively ignoring patient-specific features while focusing on disease commonality features. Our attention weight visualization analysis shows that when processing data from different patients, the model consistently assigns the highest attention weights (average 0.73) to positions containing QRS complex principal components, highly consistent with clinical practice where physicians focus on QRS morphology.

## Multidimensional in-depth analysis of performance advantages

The performance advantages of PCA-CGAN over existing methods are reflected not only in quantitative metrics but, more importantly, in the essential differences in generation quality they represent. Through systematic comparative experiments, we discovered several key patterns of performance differences:

First, from the perspective of feature fidelity of generated samples, PCA-CGAN achieves a principal component retention rate of 94.67%, far exceeding other methods (TD-GAN: 78.23%, TimeGAN: 81.45%). The root of this difference lies in the essential distinction of generation objectives. Traditional methods attempt to fit complex probability distributions in the 300-dimensional raw signal space, inevitably creating trade-offs between high-frequency details and low-frequency trends. PCA-CGAN directly models in the 16-dimensional principal component space, where each dimension corresponds to clear physiological meaning, making the generation process more controllable and stable.

Second, distribution consistency tests reveal deeper advantages. The Wasserstein distance decreased from 0.15–0.22 in other methods to 0.0847, and Jensen-Shannon divergence decreased from 0.06–0.09 to 0.0312. This order-of-magnitude improvement indicates that samples generated by PCA-CGAN are highly consistent with real data in statistical distribution. More importantly, the 2-sample KS test p-value reached 0.7834, far exceeding the statistical significance threshold, indicating that generated samples and real samples are indistinguishable in distribution. This distribution-level consistency ensures that generated samples do not introduce systematic bias, crucial for improving classifier robustness.

Third, the complete resolution of the "dilution effect" is a unique advantage of PCA-CGAN. The common problem with traditional methods is that after data augmentation, precision improves but recall decreases, forming an asymmetric performance improvement pattern. Taking SigCWGAN as an example, although average precision reaches 99.71%, average recall is only 93.28%, and for clinically critical Class A arrhythmias, recall is as low as 70.33%. In contrast, PCA-CGAN achieves simultaneous improvement in both precision (99.93%) and recall (99.88%), with Class A recall reaching 99.56%, an improvement of 29.23 percentage points.

From a machine learning theory perspective, this advantage can be explained by the "pattern averaging" problem that traditional generation methods face in optimization processes in high-dimensional spaces. When attempting to generate complete waveforms, models tend to learn the average patterns of all training samples, causing generated rare category samples to actually be closer to the feature distribution of majority categories. PCA-CGAN, by performing conditional generation in low-dimensional principal component space, provides each category with its independent generation path, avoiding this averaging effect. Mathematically, this is equivalent to decomposing the learning of joint probability distribution $P(x, y)$ into independent learning of conditional distributions $P(x|y)$, greatly reducing learning difficulty.

## In-depth analysis of technical limitations and theoretical boundaries

Despite PCA-CGAN's significant achievements, a thorough analysis of its technical limitations is crucial for understanding the method's applicability boundaries and guiding future improvements.

The primary limitation lies in PCA's inherent linear assumption. Although experiments demonstrate that linear principal components can capture over 90% of variance in ECG signals, this linear projection may not fully preserve nonlinear features of certain complex arrhythmias. For instance, some rare arrhythmias may manifest as nonlinear interactions between waveforms, and these features might be dispersed across multiple principal components in linear principal component analysis, reducing feature interpretability. Our comparative experiments show that using kernel PCA can increase the variance retention rate to 94.67%, but computational time increases 29-fold. This trade-off between efficiency and effectiveness needs to be carefully considered based on specific application scenarios.

The second limitation involves the complexity of clinical validation for generated samples. Although generated samples are highly consistent with real data in statistical metrics, the complexity of medical diagnosis requires more rigorous validation. Do the generated ECG signals contain all subtle features meaningful for diagnosis? Might they introduce artifacts that would not occur in real pathological conditions? These questions require large-scale clinical validation studies to fully answer. Our preliminary clinical evaluation shows that cardiologists' diagnostic consistency for generated samples reaches 92%, but 8% of samples are still considered to contain "atypical" features.

Third, the representativeness of the dataset limits the universality of conclusions. While the MIT-BIH dataset is a domain standard, its 47 patients are primarily from North America, with an age distribution skewed toward the elderly (average 65 years), which may not adequately represent global population diversity. Patients of different ethnicities, age groups, and comorbidity states may exhibit ECG patterns not covered by MIT-BIH. Additionally, the dataset's 360 Hz sampling rate and specific lead configuration may also affect principal component analysis results. Our sensitivity analysis indicates that when the sampling rate drops to 250 Hz, the number of required principal components increases to 20 to maintain 90% variance explanation rate.

Finally, the fixity of principal component selection may limit the model's adaptability. Our method uses a fixed 16 principal components, a choice based on statistical analysis of the MIT-BIH dataset. However, for different cardiac diseases or different clinical scenarios, the optimal number of principal components may vary. For example, applications focused on detecting subtle ST segment changes may need to retain more higher-order principal components. Developing adaptive principal component selection mechanisms that can dynamically adjust the number of principal components based on specific tasks is an important direction for future research.

These limitations do not negate the value of PCA-CGAN but rather clarify its scope of application and room for improvement. Under current technical conditions, PCA-CGAN provides an efficient and reliable solution for addressing data imbalance problems in ECG classification, particularly suitable for clinical environments requiring rapid deployment and resource constraints. Future research should gradually overcome these limitations while maintaining the method's core advantages, advancing ECG automatic diagnosis technology to higher levels.

## Conclusion

This research presents PCA-CGAN, a novel approach that fundamentally transforms ECG signal data augmentation by shifting the generation target from complete waveforms to principal component features. Through systematic experiments on the MIT-BIH dataset, we demonstrated that merely 7–16 principal components can capture over 90% of diagnostic information in ECG signals, providing theoretical justification for our paradigm shift. The proposed method successfully addresses three critical challenges in ECG classification: the extreme class imbalance leading to the "Matthew Effect," cross-patient generalization difficulties due to physiological variations, and the "dilution effect" in traditional augmentation methods. Experimental results show remarkable improvements, with the ResNet classifier achieving 99.91% average F1-score after augmentation, and most importantly, the recall for rare atrial premature beats improving from 79.58% to 99.56%. The method achieved stable convergence across 43 patients with consistent performance (NPRD: $30.6212 \pm 2.4101$), breaking the single-patient limitation that has long constrained the field.

Future research directions include extending PCA-CGAN to other biomedical signals such as EEG and EMG, where similar data imbalance challenges exist. Clinical validation studies with cardiologists are essential to ensure the diagnostic validity of generated samples and establish trust in AI-assisted diagnosis systems. Developing adaptive mechanisms for dynamic principal component selection based on specific cardiac conditions could further enhance the method's flexibility. Integration with multi-modal cardiovascular data including heart rate variability and blood pressure could provide more comprehensive diagnostic capabilities. Additionally, implementing federated learning frameworks would enable privacy-preserving training across multiple medical institutions, facilitating the development of more robust and generalizable models while addressing the critical need for larger and more diverse patient populations in ECG analysis.

## Supporting information

**S1. PCAECG_GAN.**
(RAR)

## Author contributions

**Conceptualization:** Chao Tang.

**Data curation:** Chao Tang.

**Formal analysis:** Chao Tang.

**Funding acquisition:** Chao Tang.

**Investigation:** Chao Tang.

**Methodology:** Chao Tang.

**Project administration:** Chao Tang.

**Resources:** Chao Tang.

**Software:** Chao Tang.

**Supervision:** Chao Tang.

**Validation:** Chao Tang.

**Visualization:** Chao Tang.

**Writing – original draft:** Chao Tang.

**Writing – review & editing:** Chao Tang.

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
