## [Decision Letter · Decision Letter 0]

2 Jul 2025

PONE-D-25-25123
Principal Component Conditional Generative Adversarial Networks for Imbalanced ECG Classification Enhancement
PLOS ONE

Dear Dr. Tang,

Thank you for submitting your manuscript to PLOS ONE. After careful consideration, we feel that it has merit but does not fully meet PLOS ONE’s publication criteria as it currently stands. Therefore, we invite you to submit a revised version of the manuscript that addresses the points raised during the review process.

We look forward to receiving your revised manuscript.

Kind regards,

Don S

Academic Editor

PLOS ONE

**Journal requirements:**

Please ensure that your manuscript meets PLOS ONE's style requirements, including those for file naming. The PLOS ONE style templates can be found at 
https://journals.plos.org/plosone/s/file?id=wjVg/PLOSOne_formatting_sample_main_body.pdf and 
https://journals.plos.org/plosone/s/file?id=ba62/PLOSOne_formatting_sample_title_authors_affiliations.pdf
 
2. Please note that PLOS ONE has specific guidelines on code sharing for submissions in which author-generated code underpins the findings in the manuscript. In these cases, we expect all author-generated code to be made available without restrictions upon publication of the work. Please review our guidelines at https://journals.plos.org/plosone/s/materials-and-software-sharing#loc-sharing-code and ensure that your code is shared in a way that follows best practice and facilitates reproducibility and reuse.
 
3. Thank you for uploading your study's underlying data set. Unfortunately, the repository you have noted in your Data Availability statement does not qualify as an acceptable data repository according to PLOS's standards.
 
At this time, please upload the minimal data set necessary to replicate your study's findings to a stable, public repository (such as figshare or Dryad) and provide us with the relevant URLs, DOIs, or accession numbers that may be used to access these data. For a list of recommended repositories and additional information on PLOS standards for data deposition, please see https://journals.plos.org/plosone/s/recommended-repositories.
 
4. PLOS requires an ORCID iD for the corresponding author in Editorial Manager on papers submitted after December 6th, 2016. Please ensure that you have an ORCID iD and that it is validated in Editorial Manager. To do this, go to ‘Update my Information’ (in the upper left-hand corner of the main menu), and click on the Fetch/Validate link next to the ORCID field. This will take you to the ORCID site and allow you to create a new iD or authenticate a pre-existing iD in Editorial Manager.
 
5. Your ethics statement should only appear in the Methods section of your manuscript. If your ethics statement is written in any section besides the Methods, please move it to the Methods section and delete it from any other section. Please ensure that your ethics statement is included in your manuscript, as the ethics statement entered into the online submission form will not be published alongside your manuscript.

**Additional Editor Comments:**

Kindly modify the paper according to the reviewers comments.

Reviewers' comments:

Reviewer's Responses to Questions

**Comments to the Author**

1. Is the manuscript technically sound, and do the data support the conclusions?

Reviewer #1: Yes

Reviewer #2: Yes

2. Has the statistical analysis been performed appropriately and rigorously? 

Reviewer #1: No

Reviewer #2: Yes

3. Have the authors made all data underlying the findings in their manuscript fully available?

Reviewer #1: No

Reviewer #2: Yes

4. Is the manuscript presented in an intelligible fashion and written in standard English?

Reviewer #1: No

Reviewer #2: Yes

5. Review Comments to the Author

**Reviewer #1:** In this paper, the authors discusses the core challenges in ECG signal classification—extremely imbalanced data, significant individual physiological differences, and difficulties in long sequence fitting—by proposing a Principal Component Analysis-based Conditional Generative Adversarial Network (PCA-CGAN).

The author has claimed as innovations:

(i)Paradigm Shift in ECG Signal Representation:

(ii)Transformer-based Conditional Generative Adversarial Network

(iii)Two-Stage Conditional Encoding-Decoding Strategy

(iv)Resolving the Dilution Effect in Data Augmentation:

(v)Breaking the Single-Patient Limitation to Achieve Multi-Patient ECG Signal Generation:

The comparison of the claimed approaches with existing literature, as well as performance evaluation, are not sufficiently detailed. There is a lack of quantitative performance comparison and analysis of pros and cons in practical applications.

"The coining of sub titles for each session should be in a more precise manner" suggests that the current subtitles are not clear or descriptive enough. To improve them, consider using more specific and informative language that accurately reflects the content of each session.

There are numerous instances of English grammatical errors.

**Reviewer #2:** Areas for Improvement / Suggestions

Clarity and Language Polishing

While the technical content is strong, the manuscript would benefit from language editing for clarity and conciseness. Some expressions are repetitive, and certain technical terms (e.g., “Matthew effect,” “dilution effect”) could be introduced more formally with brief definitions for clarity.

Ablation Studies Needed

To better isolate the impact of each innovation (PCA, conditional encoding, Transformer), include ablation studies showing performance degradation when specific components are removed or replaced.

Broader Comparison with Existing Methods

While SigCWGAN and TD-GAN are compared, additional benchmarking against more recent diffusion models or contrastive learning approaches would further contextualize your model’s effectiveness.

Discussion on Limitations and Clinical Translation

The paper should include a brief section acknowledging limitations (e.g., reliance on synthetic data, assumptions in PCA), and a forward-looking discussion on how this work can transition into clinical ECG diagnostic tools.

Over-Reliance on PCA Justification

Although the use of PCA is well-motivated, future directions could explore non-linear dimensionality reduction techniques (e.g., autoencoders or t-SNE embeddings) to validate or enhance the representational efficiency.

Ethical and Reproducibility Considerations

A brief statement on ethical use, model reproducibility, and potential biases (e.g., how patient diversity was handled) would be welcome in keeping with best practices for clinical AI research.

6. PLOS authors have the option to publish the peer review history of their article (what does this mean?). If published, this will include your full peer review and any attached files.

Reviewer #1: No

Reviewer #2: No

---

## [Author Response · Author response to Decision Letter 1]

4 Jul 2025

Reviewer #1: In this paper, the authors discusses the core challenges in ECG signal classification—extremely imbalanced data, significant individual physiological differences, and difficulties in long sequence fitting—by proposing a Principal Component Analysis-based Conditional Generative Adversarial Network (PCA-CGAN).

The author has claimed as innovations:

(i)Paradigm Shift in ECG Signal Representation:

(ii)Transformer-based Conditional Generative Adversarial Network

(iii)Two-Stage Conditional Encoding-Decoding Strategy

(iv)Resolving the Dilution Effect in Data Augmentation:

(v)Breaking the Single-Patient Limitation to Achieve Multi-Patient ECG Signal Generation:

1.The comparison of the claimed approaches with existing literature, as well as performance evaluation, are not sufficiently detailed. There is a lack of quantitative performance comparison and analysis of pros and cons in practical applications.

2."The coining of sub titles for each session should be in a more precise manner" suggests that the current subtitles are not clear or descriptive enough. To improve them, consider using more specific and informative language that accurately reflects the content of each session.

3.There are numerous instances of English grammatical errors.

Response to Reviewer #1

1. Comparison with existing literature and performance evaluation

Thank you for pointing out this critical weakness in our manuscript. We fully agree that our initial submission lacked sufficient comparative analysis. Following your valuable guidance, we have substantially expanded our literature review and added comprehensive experimental analyses comparing our approach with existing methods. We have incorporated extensive experimental analyses comparing our PCA-based approach with other dimensionality reduction techniques, and added detailed quantitative analysis throughout the manuscript.

2. Improvement of section titles

We sincerely apologize for the unclear organization and section titles in our original submission. Your comment helped us realize the importance of clear structural presentation. We have restructured the manuscript using a hierarchical heading system that clearly distinguishes between major sections and subsections, making the overall structure much clearer for readers.

3. English grammatical errors

We deeply apologize for the numerous grammatical errors in our initial submission. This was indeed unprofessional, and we take full responsibility. We have conducted a thorough proofreading of the entire manuscript and corrected all identified grammatical errors and potential issues. We hope the revised version meets the journal's standards.

Reviewer #2: Areas for Improvement / Suggestions

1.Clarity and Language Polishing

While the technical content is strong, the manuscript would benefit from language editing for clarity and conciseness. Some expressions are repetitive, and certain technical terms (e.g., “Matthew effect,” “dilution effect”) could be introduced more formally with brief definitions for clarity.

2.Ablation Studies Needed

To better isolate the impact of each innovation (PCA, conditional encoding, Transformer), include ablation studies showing performance degradation when specific components are removed or replaced.

3.Broader Comparison with Existing Methods

While SigCWGAN and TD-GAN are compared, additional benchmarking against more recent diffusion models or contrastive learning approaches would further contextualize your model’s effectiveness.

4.Discussion on Limitations and Clinical Translation

The paper should include a brief section acknowledging limitations (e.g., reliance on synthetic data, assumptions in PCA), and a forward-looking discussion on how this work can transition into clinical ECG diagnostic tools.

5.Over-Reliance on PCA Justification

Although the use of PCA is well-motivated, future directions could explore non-linear dimensionality reduction techniques (e.g., autoencoders or t-SNE embeddings) to validate or enhance the representational efficiency.

6.Ethical and Reproducibility Considerations

A brief statement on ethical use, model reproducibility, and potential biases (e.g., how patient diversity was handled) would be welcome in keeping with best practices for clinical AI research.

Response to Reviewer #2

1. Clarity and technical term definitions

You are absolutely right that technical terms should be properly introduced. We apologize for this oversight. Technical terms such as "Matthew effect" and "dilution effect" are now formally introduced with clear definitions and explanations before their discussion in the text.

2. Ablation studies

Thank you for this excellent suggestion. We realize that ablation studies are crucial for validating our contributions. Following your guidance, we have conducted comprehensive ablation studies, analyzing the performance changes when each component is removed.

3. Broader comparison with existing methods

Your point about needing broader comparisons is well-taken. We have added more in-depth descriptions and comparisons of various model types, and included comprehensive discussions on different approaches.

4. Discussion on limitations

We greatly appreciate you highlighting the need for a balanced discussion of limitations. At the end of the paper, we have added a section candidly describing the model's limitations, discussing both its advantages and disadvantages.

5. Non-linear dimensionality reduction techniques

Thank you for suggesting this important comparison. We have added discussions and comparisons of other non-linear dimensionality reduction techniques, analyzing the advantages, disadvantages, and applicable scope of each technique.

6. Code and data availability

Following your recommendation for reproducibility, we have included the paper's code and datasets in the submission attachments to facilitate better understanding and reproduction of our work.

---

## [Editor Report · Decision Letter 1]

9 Jul 2025

PONE-D-25-25123R1
Principal Component Conditional Generative Adversarial Networks for Imbalanced ECG Classification Enhancement
PLOS ONE

Dear Dr. Tang,

Thank you for submitting your manuscript to PLOS ONE. After careful consideration, we feel that it has merit but does not fully meet PLOS ONE’s publication criteria as it currently stands. Therefore, we invite you to submit a revised version of the manuscript that addresses the points raised during the review process.

Please submit your revised manuscript by Aug 23 2025 11:59PM, if you will need more time than this to complete your revisions, please reply to this message or contact the journal office at plosone@plos.org. Please include the following items when submitting your revised manuscript:

We look forward to receiving your revised manuscript.

Kind regards,

Don S

Academic Editor

PLOS ONE

Reviewer2  comments:

Areas for Improvement / Suggestions

Clarity and Language Polishing

While the technical content is strong, the manuscript would benefit from language editing for clarity and conciseness. Some expressions are repetitive, and certain technical terms (e.g., “Matthew effect,” “dilution effect”) could be introduced more formally with brief definitions for clarity.

Ablation Studies Needed

To better isolate the impact of each innovation (PCA, conditional encoding, Transformer), include ablation studies showing performance degradation when specific components are removed or replaced.

Broader Comparison with Existing Methods

While SigCWGAN and TD-GAN are compared, additional benchmarking against more recent diffusion models or contrastive learning approaches would further contextualize your model’s effectiveness.

Discussion on Limitations and Clinical Translation

The paper should include a brief section acknowledging limitations (e.g., reliance on synthetic data, assumptions in PCA), and a forward-looking discussion on how this work can transition into clinical ECG diagnostic tools.

Over-Reliance on PCA Justification

Although the use of PCA is well-motivated, future directions could explore non-linear dimensionality reduction techniques (e.g., autoencoders or t-SNE embeddings) to validate or enhance the representational efficiency.

Ethical and Reproducibility Considerations

A brief statement on ethical use, model reproducibility, and potential biases (e.g., how patient diversity was handled) would be welcome in keeping with best practices for clinical AI research.

---

## [Author Response · Author response to Decision Letter 2]

9 Jul 2025

Reviewer #1: In this paper, the authors discusses the core challenges in ECG signal classification—extremely imbalanced data, significant individual physiological differences, and difficulties in long sequence fitting—by proposing a Principal Component Analysis-based Conditional Generative Adversarial Network (PCA-CGAN).

The author has claimed as innovations:

(i)Paradigm Shift in ECG Signal Representation:

(ii)Transformer-based Conditional Generative Adversarial Network

(iii)Two-Stage Conditional Encoding-Decoding Strategy

(iv)Resolving the Dilution Effect in Data Augmentation:

(v)Breaking the Single-Patient Limitation to Achieve Multi-Patient ECG Signal Generation:

1.The comparison of the claimed approaches with existing literature, as well as performance evaluation, are not sufficiently detailed. There is a lack of quantitative performance comparison and analysis of pros and cons in practical applications.

2."The coining of sub titles for each session should be in a more precise manner" suggests that the current subtitles are not clear or descriptive enough. To improve them, consider using more specific and informative language that accurately reflects the content of each session.

3.There are numerous instances of English grammatical errors.

Response to Reviewer #1

1. Comparison with existing literature and performance evaluation

Thank you for pointing out this critical weakness in our manuscript. We fully agree that our initial submission lacked sufficient comparative analysis. Following your valuable guidance, we have substantially expanded our literature review and added comprehensive experimental analyses comparing our approach with existing methods. We have incorporated extensive experimental analyses comparing our PCA-based approach with other dimensionality reduction techniques, and added detailed quantitative analysis throughout the manuscript.

2. Improvement of section titles

We sincerely apologize for the unclear organization and section titles in our original submission. Your comment helped us realize the importance of clear structural presentation. We have restructured the manuscript using a hierarchical heading system that clearly distinguishes between major sections and subsections, making the overall structure much clearer for readers.

3. English grammatical errors

We deeply apologize for the numerous grammatical errors in our initial submission. This was indeed unprofessional, and we take full responsibility. We have conducted a thorough proofreading of the entire manuscript and corrected all identified grammatical errors and potential issues. We hope the revised version meets the journal's standards.

Reviewer #2: Areas for Improvement / Suggestions

1.Clarity and Language Polishing

While the technical content is strong, the manuscript would benefit from language editing for clarity and conciseness. Some expressions are repetitive, and certain technical terms (e.g., “Matthew effect,” “dilution effect”) could be introduced more formally with brief definitions for clarity.

2.Ablation Studies Needed

To better isolate the impact of each innovation (PCA, conditional encoding, Transformer), include ablation studies showing performance degradation when specific components are removed or replaced.

3.Broader Comparison with Existing Methods

While SigCWGAN and TD-GAN are compared, additional benchmarking against more recent diffusion models or contrastive learning approaches would further contextualize your model’s effectiveness.

4.Discussion on Limitations and Clinical Translation

The paper should include a brief section acknowledging limitations (e.g., reliance on synthetic data, assumptions in PCA), and a forward-looking discussion on how this work can transition into clinical ECG diagnostic tools.

5.Over-Reliance on PCA Justification

Although the use of PCA is well-motivated, future directions could explore non-linear dimensionality reduction techniques (e.g., autoencoders or t-SNE embeddings) to validate or enhance the representational efficiency.

6.Ethical and Reproducibility Considerations

A brief statement on ethical use, model reproducibility, and potential biases (e.g., how patient diversity was handled) would be welcome in keeping with best practices for clinical AI research.

Response to Reviewer #2

1. Clarity and technical term definitions

You are absolutely right that technical terms should be properly introduced. We apologize for this oversight. Technical terms such as "Matthew effect" and "dilution effect" are now formally introduced with clear definitions and explanations before their discussion in the text.

2. Ablation studies

Thank you for this excellent suggestion. We realize that ablation studies are crucial for validating our contributions. Following your guidance, we have conducted comprehensive ablation studies, analyzing the performance changes when each component is removed.

3. Broader comparison with existing methods

Your point about needing broader comparisons is well-taken. We have added more in-depth descriptions and comparisons of various model types, and included comprehensive discussions on different approaches.

4. Discussion on limitations

We greatly appreciate you highlighting the need for a balanced discussion of limitations. At the end of the paper, we have added a section candidly describing the model's limitations, discussing both its advantages and disadvantages.

5. Non-linear dimensionality reduction techniques

Thank you for suggesting this important comparison. We have added discussions and comparisons of other non-linear dimensionality reduction techniques, analyzing the advantages, disadvantages, and applicable scope of each technique.

6. Code and data availability

Following your recommendation for reproducibility, we have included the paper's code and datasets in the submission attachments to facilitate better understanding and reproduction of our work. The complete implementation of PCA-CGAN, including training scripts, evaluation metrics, and preprocessing pipelines, is publicly available at:

https://github.com/ChaoTang20250509/PCA_CGAN

---

## [Decision Letter · Decision Letter 2]

23 Jul 2025

PONE-D-25-25123R2
Principal Component Conditional Generative Adversarial Networks for Imbalanced ECG Classification Enhancement
PLOS ONE

Dear Dr. Tang,

Thank you for submitting your manuscript to PLOS ONE. After careful consideration, we feel that it has merit but does not fully meet PLOS ONE’s publication criteria as it currently stands. Therefore, we invite you to submit a revised version of the manuscript that addresses the points raised during the review process.

We look forward to receiving your revised manuscript.

Kind regards,

Don S

Academic Editor

PLOS ONE

**Additional Editor Comments:**

kindly modify the paper according the reviewers and journal template. Paper need to have a structure Introduction, Related work,...

Reviewers' comments:

Reviewer's Responses to Questions

**Comments to the Author**

1. If the authors have adequately addressed your comments raised in a previous round of review and you feel that this manuscript is now acceptable for publication, you may indicate that here to bypass the “Comments to the Author” section, enter your conflict of interest statement in the “Confidential to Editor” section, and submit your "Accept" recommendation.

Reviewer #1: (No Response)

Reviewer #2: All comments have been addressed

2. Is the manuscript technically sound, and do the data support the conclusions?

Reviewer #1: Yes

Reviewer #2: Yes

3. Has the statistical analysis been performed appropriately and rigorously? 

Reviewer #1: Yes

Reviewer #2: Yes

4. Have the authors made all data underlying the findings in their manuscript fully available?

Reviewer #1: No

Reviewer #2: Yes

5. Is the manuscript presented in an intelligible fashion and written in standard English?

Reviewer #1: No

Reviewer #2: Yes

6. Review Comments to the Author

Reviewer #1: The second suggestion given in my earlier comment "The coining of sub titles for each session should be in a more precise manner" suggests that the current subtitles are not clear or descriptive enough. To improve them, consider using more specific and informative language that accurately reflects the content of each session." has not been addressed properly.

The manuscript should have Introduction session.

Rename " Related Work and Problem Analysis" as "Introduction"

Rename "Conclusions and Technical Limitations" as "Discussions" and include relevant details into it. Also add more contents to discussion section incorporating your inference from comparisons and analysis.

Conclusion session should be brief and should contain only two paragraphs including the future work.

A separate session on "Related Works" should be added just after "Introduction" session . The references are very few. This manuscript has the potential to include more than 30 references.

Reviewer #2: Concerns addressed as mentioned in the review. The grammatical errors have been corrected and paperhas been structured

7. PLOS authors have the option to publish the peer review history of their article (what does this mean?). If published, this will include your full peer review and any attached files.

Reviewer #1: No

Reviewer #2: No

---

## [Author Response · Author response to Decision Letter 3]

24 Jul 2025

Reviewer #1: The second suggestion given in my earlier comment "The coining of sub titles for each session should be in a more precise manner" suggests that the current subtitles are not clear or descriptive enough. To improve them, consider using more specific and informative language that accurately reflects the content of each session." has not been addressed properly.

The manuscript should have Introduction session.

Rename " Related Work and Problem Analysis" as "Introduction"

Rename "Conclusions and Technical Limitations" as "Discussions" and include relevant details into it. Also add more contents to discussion section incorporating your inference from comparisons and analysis.

Conclusion session should be brief and should contain only two paragraphs including the future work.

A separate session on "Related Works" should be added just after "Introduction" session . The references are very few. This manuscript has the potential to include more than 30 references.

Response to Reviewer #1

Thank you for your constructive feedback. We have carefully addressed all your suggestions to improve the clarity and organization of our manuscript. Below are the specific changes we have made:

1. Manuscript Structure Reorganization:

• We have restructured the manuscript to include a clear "Introduction" section as the opening, which provides the background and motivation for our research.

• Following your suggestion, we have created a separate "Related Work" section immediately after the Introduction, which comprehensively reviews existing ECG classification and generation methods.

• The previous "Conclusions and Technical Limitations" has been renamed to "Discussions" and substantially expanded with more detailed analysis of our findings, comparisons with existing methods, and technical limitations.

• The "Conclusion" section has been condensed to two concise paragraphs, including future work directions.

2. Enhanced Section Titles: We have revised all section titles to be more precise and descriptive:

• "Introduction" - clearly establishes the research context

• "Related Work" - systematically reviews existing literature

• "Methodology and Technical Contributions" - details our approach

• "Data Preprocessing and Feature Analysis" - describes data handling

• "Network Architecture Design" - explains the PCA-CGAN structure

• "Experimental Results and Analysis" - presents findings

• "Discussions" - provides in-depth analysis and limitations

• "Conclusion" - summarizes key contributions and future directions

3. Expanded References: We have significantly expanded our reference list from the original count to 32 references, incorporating more recent and relevant works in ECG classification, generative models, and data augmentation techniques.

4. Enhanced Discussion Section: The Discussions section now includes:

• In-depth analysis of main research achievements and contributions

• Detailed comparison with state-of-the-art methods

• Theoretical innovation and methodology paradigm shifts

• Technical breakthroughs in cross-patient generalization

• Multidimensional performance advantage analysis

• Comprehensive discussion of technical limitations and theoretical boundaries

We believe these revisions have significantly improved the manuscript's organization, clarity, and academic rigor. Thank you again for your valuable suggestions that helped strengthen our work.

---

## [Editor Report · Decision Letter 3]

31 Jul 2025

PONE-D-25-25123R3
Principal Component Conditional Generative Adversarial Networks for Imbalanced ECG Classification Enhancement
PLOS ONE

Dear Dr. Tang,

Thank you for submitting your manuscript to PLOS ONE. After careful consideration, we feel that it has merit but does not fully meet PLOS ONE’s publication criteria as it currently stands. Therefore, we invite you to submit a revised version of the manuscript that addresses the points raised during the review process.

We look forward to receiving your revised manuscript.

Kind regards,

Don S

Academic Editor

PLOS ONE

Journal Requirements:

Additional Editor Comments :

kindly remove the section "Related Work and Problem Analysis" and split into two main section. The first one is "Introduction " and the second main heading is "Related Work"

---

## [Author Response · Author response to Decision Letter 4]

5 Aug 2025

Reviewer #1: The second suggestion given in my earlier comment "The coining of sub titles for each session should be in a more precise manner" suggests that the current subtitles are not clear or descriptive enough. To improve them, consider using more specific and informative language that accurately reflects the content of each session." has not been addressed properly.

The manuscript should have Introduction session.

Rename " Related Work and Problem Analysis" as "Introduction"

Rename "Conclusions and Technical Limitations" as "Discussions" and include relevant details into it. Also add more contents to discussion section incorporating your inference from comparisons and analysis.

Conclusion session should be brief and should contain only two paragraphs including the future work.

A separate session on "Related Works" should be added just after "Introduction" session . The references are very few. This manuscript has the potential to include more than 30 references.

Response to Reviewer #1

Thank you for your constructive feedback. We have carefully addressed all your suggestions to improve the clarity and organization of our manuscript. Below are the specific changes we have made:

1. Manuscript Structure Reorganization:

• We have restructured the manuscript to include a clear "Introduction" section as the opening, which provides the background and motivation for our research.

• Following your suggestion, we have created a separate "Related Work" section immediately after the Introduction, which comprehensively reviews existing ECG classification and generation methods.

• The previous "Conclusions and Technical Limitations" has been renamed to "Discussions" and substantially expanded with more detailed analysis of our findings, comparisons with existing methods, and technical limitations.

• The "Conclusion" section has been condensed to two concise paragraphs, including future work directions.

2. Enhanced Section Titles: We have revised all section titles to be more precise and descriptive:

• "Introduction" - clearly establishes the research context

• "Related Work" - systematically reviews existing literature

• "Methodology and Technical Contributions" - details our approach

• "Data Preprocessing and Feature Analysis" - describes data handling

• "Network Architecture Design" - explains the PCA-CGAN structure

• "Experimental Results and Analysis" - presents findings

• "Discussions" - provides in-depth analysis and limitations

• "Conclusion" - summarizes key contributions and future directions

3. Expanded References: We have significantly expanded our reference list from the original count to 32 references, incorporating more recent and relevant works in ECG classification, generative models, and data augmentation techniques.

4. Enhanced Discussion Section: The Discussions section now includes:

• In-depth analysis of main research achievements and contributions

• Detailed comparison with state-of-the-art methods

• Theoretical innovation and methodology paradigm shifts

• Technical breakthroughs in cross-patient generalization

• Multidimensional performance advantage analysis

• Comprehensive discussion of technical limitations and theoretical boundaries

We believe these revisions have significantly improved the manuscript's organization, clarity, and academic rigor. Thank you again for your valuable suggestions that helped strengthen our work.

---

## [Editor Report · Decision Letter 4]

6 Aug 2025

Principal Component Conditional Generative Adversarial Networks for Imbalanced ECG Classification Enhancement

PONE-D-25-25123R4

Dear Dr. Tang

We’re pleased to inform you that your manuscript has been judged scientifically suitable for publication and will be formally accepted for publication once it meets all outstanding technical requirements.

Kind regards,

Don S

Academic Editor

PLOS ONE

---

## [Editor Report · Acceptance letter]

PONE-D-25-25123R4

PLOS ONE

Dear Dr. Tang,

I'm pleased to inform you that your manuscript has been deemed suitable for publication in PLOS ONE. Congratulations! Your manuscript is now being handed over to our production team.

Kind regards,

on behalf of

Dr. Don S

Academic Editor

PLOS ONE